# RNA G-quadruplex in TMPRSS2 reduces SARS-CoV-2 infection

Geng Liu[1,6], Wenya Du[1,6], Xiongbo Sang[1], Qiyu Tong[1], Ye Wang[2], Guoqing Chen[3], Yi Yuan[4], Lili Jiang[5], Wei Cheng [3], Dan Liu[2], Yan Tian[1] & Xianghui Fu [1✉]

Severe acute respiratory syndrome coronavirus 2 (SARS-CoV-2) infection continues to have devastating consequences worldwide. Recently, great efforts have been made to identify SARS-CoV-2 host factors, but the regulatory mechanisms of these host molecules, as well as the virus per se, remain elusive. Here we report a role of RNA G-quadruplex (RG4) in SARS-CoV-2 infection. Combining bioinformatics, biochemical and biophysical assays, we demonstrate the presence of RG4s in both SARS-CoV-2 genome and host factors. The biological and pathological importance of these RG4s is then exemplified by a canonical 3-quartet RG4 within *Tmprss2*, which can inhibit *Tmprss2* translation and prevent SARS-CoV-2 entry. Intriguingly, G-quadruplex (G4)-specific stabilizers attenuate SARS-CoV-2 infection in pseudovirus cell systems and mouse models. Consistently, the protein level of TMPRSS2 is increased in lungs of COVID-19 patients. Our findings reveal a previously unknown mechanism underlying SARS-CoV-2 infection and suggest RG4 as a potential target for COVID-19 prevention and treatment.

[1] Division of Endocrinology and Metabolism, National Clinical Research Center for Geriatrics, State Key Laboratory of Biotherapy and Cancer Center, West China Hospital, Sichuan University and Collaborative Innovation Center of Biotherapy, Chengdu 610041 Sichuan, China. [2] Division of Pulmonary and Critical Care Medicine, West China Hospital, Sichuan University, Chengdu 610041 Sichuan, China. [3] Division of Respiratory and Critical Care Medicine, State Key Laboratory of Biotherapy and Cancer Center, West China Hospital, Sichuan University, Chengdu 610041 Sichuan, China. [4] Natural Products Research Center, Chengdu Institute of Biology, Chinese Academy of Sciences, Chengdu 610041 Sichuan, China. [5] Lab of Pathology, West China Hospital, Sichuan University, Chengdu 610041 Sichuan, China. [6] These authors contributed equally: Geng Liu, Wenya Du. ✉email: xfu@scu.edu.cn

The ongoing pandemic of coronavirus disease 2019 (COVID-19), caused by the novel severe acute respiratory syndrome coronavirus 2 (SARS-CoV-2), has rapidly spread worldwide and aroused global healthy, social, and economic disruption[1,2]. It has led to >250 million diagnosed cases and >5 million deaths in worldwide (by WHO), and the number of new infections and deaths is still increasing alarmingly. Moreover, the fast-evolved variation of SARS-CoV-2, which is more transmissible and spread globally compared with the ancestral virus, adds another layer of complexity and uncertainty into current therapeutic strategies for this deadly virus[3–5]. Although the global scientific community has been racing to find treatments that could bring the pandemic under control, there are no effective target drugs against the SARS-CoV-2 virus to date. Under these circumstances, better understanding the SARS-CoV-2 pathogenesis and its extended implications to bring new concepts for antiviral treatments are becoming urgent and significant.

SARS-CoV-2 contains a single-stranded positive-sense RNA genome that encodes 16 non-structural proteins (NSPs), several accessory proteins, and four essential structural proteins, including small envelope (E) protein, nucleocapsid (N) protein, spike (S) glycoprotein, and matrix (M) protein[6]. The interaction between the virus and its host is central to completing the virus lifecycle, including virus entry, replication, and dissemination. Specifically, angiotensin-converting enzyme 2 (ACE2) and transmembrane serine protease 2 (TMPRSS2)[7] have been regarded as two essential host determinants for SARS-CoV-2 infection and pathogenesis. When SARS-CoV-2 invades, the S glycoprotein initially engages ACE2 as a cellular receptor for entry and then is enzymatically activated by TMPRSS2, which is essential for efficient fusion and release of the virus contents into the host cell cytosol[7]. Interestingly, expression of ACE2 and TMPRSS2, as well as other host factors, was found to be responsive to various cellular conditions related to COVID-19[7], suggesting a potential association of their expression with SARS-CoV-2 infection and pathogenesis. Inspired by their pathologic importance, certain inhibitors against SARS-CoV-2 host factors have been developed to combat COVID-19 pandemic. For instance, camostat mesylate, an orally available well-known TMPRSS2 inhibitor, has been hypothesized as a potential antiviral drug against COVID-19 and is currently ongoing clinical trials[8]. However, the mechanisms underlying the regulation of these host factors in COVID-19 etiology remain unknown.

The genome of positive-sense RNA virus, including SARS-CoV-2, also serves as the template for both replication and translation and thus is rich in functional RNA elements that can modulate the gene regulation machinery even before the accumulation of viral proteins. RNA G-quadruplex (RG4) is a non-canonical RNA secondary structure formed by guanine (G)-rich sequences, which contains two or more layers of G-quartets assembled in a planar arrangement by Hoogsteen hydrogen bonding[9,10]. RG4 is enriched in numerous viruses, including ebola virus[11], hepatitis C virus[12], human immunodeficiency virus[13], and SARS-CoV[14], and has been implicated in virus pathogenesis and infectious diseases. Moreover, recent RG4 profiling analyses have identified thousands of putative RG4 regions in human transcriptome[15,16], strongly suggesting a causal role of this structural RNA element in physiology and pathology. Indeed, RG4 can regulate various biological processes[17–21] and participate in numerous human diseases, such as cancer[18,22], neurological diseases[23], metabolic disorders[24], and viral diseases[25], and thus is emerging as a therapeutic target. Interestingly, RG4s within cellular host mRNAs may interact with the NSP3 of SARS-CoV, thereby interfering with host cell antiviral response[14]. Given the widespread distribution and function of RG4 in both virus and the human genome, together with recent clues[26–28], it is of particular interest to explore the potential involvement of RG4 in the extreme pathogenicity of SARS-CoV-2.

Herein, we identify the role of RG4 in SARS-CoV-2 infection and pathogenesis. The presence of RG4 in both SARS-CoV-2 genome and host factors are characterized by a combination of multiple bioinformatics, biochemical, and biophysical assays. The biological and pathological importance of RG4 in SARS-CoV-2 pathogenesis is then exemplified by a single RG4 in *Tmprss2*. This canonical 3-quartet RG4 structure in the open reading frame (ORF) region can inhibit *Tmprss2* translation. Correspondingly, G-quadruplex (G4)-specific stabilizers attenuate SARS-CoV-2 entry to host cells in both pseudovirus cell systems and mouse models. These findings uncover a mechanism underlying SARS-CoV-2 infection and suggest the potential of RG4-targeting strategies for COVID-19 prevention.

## Results

**Identification of RG4s in SARS-CoV-2 genome**. To explore the potential involvement of RG4 in COVID-19, we initially investigated putative G4-forming sequences (PQSs) in SARS-CoV-2 genome via in-house analysis and adopted several methods to verify RG4 formation. 15 PQSs were predicted in SARS-CoV-2 genome by QGRS-mapper algorithm[29] (Supplementary Fig. 1a and Supplementary Table 1). These regions all fit the $G_2N_{1-7}G_2N_{1-7}G_2N_{1-7}G_2$ formula, indicating the potential to adopt the non-canonical and metastable RG4 structures with 2-quartet[26,27].

Next, the top two PQSs (PQS-13385 and PQS-24268) were chosen for experimental verification by using circular dichroism (CD) and fluorescence emission spectrum measurements. Under 150 mM KCl condition that is a prerequisite for RG4 formation in vitro, the CD spectrum of PQS-13385 exhibited a positive CD peak ~264 nm and a negative band close to 238 nm (Supplementary Fig. 1b), which is in line with the characteristics of a 2-quartet RG4 topology[30]. *N*-methyl mesophorphyrin IX (NMM) and thioflavin T (ThT), two reliable G4-targeted fluorescence light-up probes[31–33], enable quantitative measurement of RG4 folding propensity and conformation specificity. Notably, the fluorescence of NMM and ThT for PQS-13385 were increased in the presence of KCl (Supplementary Fig. 1b). In contrast, mutated guanines (Gs) in G-tracts of PQS-13385 abolished these observations (Supplementary Fig. 1b), further supporting the capability of this PQS for RG4 formation. Unlike PQS-13385, PQS-24268 did not display classic RG4 features, as evidenced by CD spectrum and fluorescence probes (Supplementary Fig. 1b). CD-melting analysis revealed that the melting temperature (Tm) of PQS-13385 was increased in the presence of physiologically relevant levels of KCl (150 mM) as compared with LiCl (Supplementary Fig. 1c), indicative of a thermostable RG4 structure.

Together, these results demonstrate the presence of RG4 in SARS-CoV-2 genome, as exemplified by PQS-13385 embedded in the coding sequences region of NSP10, suggesting a possible role of RG4 in SARS-COV-2 pathogenesis.

**Characterization of RG4s in SARS-CoV-2 host factors**. Similar characterization of RG4 in SARS-CoV-2 genome has been reported by three recent studies[26–28]. More importantly, it appears to be impracticable to determine the functional impacts of RG4 within SARS-CoV-2 genome in virus infection and pathogenesis. Therefore, we extended the analysis of RG4 to SARS-CoV-2 host factors. *Ace2* and *Tmprss2*, the two most common cellular entry determinants for SARS-CoV-2[7], were predicted to have multiple PQSs with considerable variation in evolutionary conservation (Fig. 1a and Supplementary Table 2). PQSs in *Ace2* showed the potential to adopt the non-canonical 2-quartet RG4 structures (Supplementary Fig. 1f and Supplementary Table 2). *Tmprss2* PQSs had higher G-scores than those of *Ace2* (Fig. 1a) and SARS-CoV-2 genome (Supplementary Fig. 1a), indicating increased probability for RG4 formation. Among the

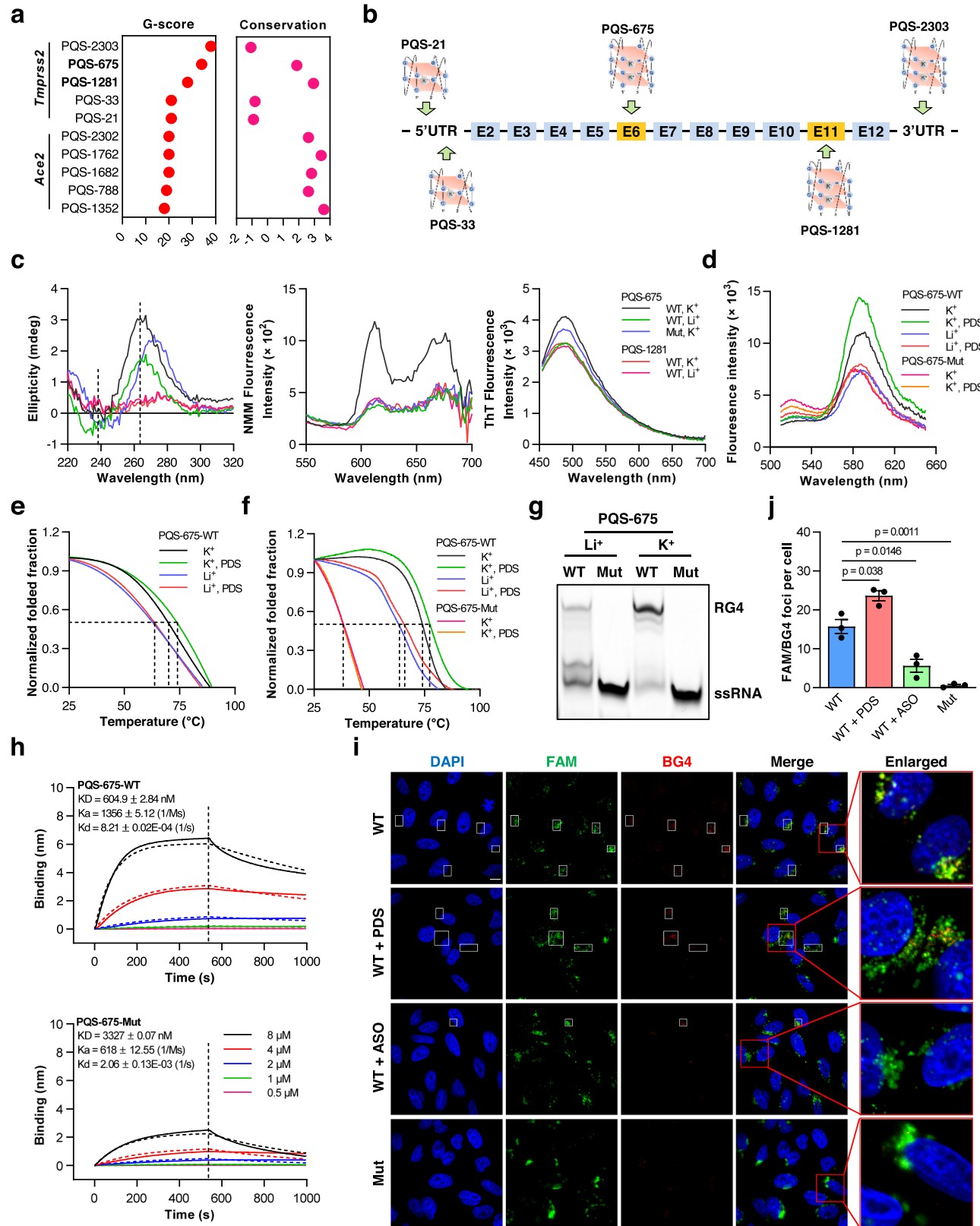

top five PQSs in *Tmprss2*, PQS-675, PQS-1281, and PQS-2303 were predicted to form the canonical 3-quartet RG4 structure (Fig. 1b), which is more stable than 2-quartet RG4. In addition, PQS-675 and PQS-1281 are located in the ORF of *Tmprss2* and are highly conserved among vertebrates (Fig. 1a, b and Supplementary Fig. 1e), implying potential significance in TMPRSS2

regulation. Thus, PQS-675 and PQS-1281 were used to exemplify the existence of RG4 in SARS-CoV-2 host factors.

Under 150 mM KCl condition, the CD spectrum of PQS-675, but not PQS-1281, showed typical positive and negative molar ellipticity peaks (Fig. 1c). The fluorescence emission spectrum further confirmed the ability of PQS-675, but not PQS-1281, to

**Fig. 1 Characterization of RG4 in human _Tmprss2_. a** RG4 potential and sequence conservation of _Tmprss2_ and _Ace2_ mRNA. The PQSs are predicted by QGRS-mapper. The conservation of the PQSs is assessed by UCSC genome browser. **b** Five top PQS sites in _Tmprss2_ mRNA. **c** CD spectrum (left panel), NMM (middle panel), and ThT (right panel) fluorescence emission spectra of PQS-675-WT, PQS-675-Mut, and PQS-1281-WT RNA under KCl or LiCl conditions. **d** FRET spectrum of PQS-675-WT and PQS-675-Mut RNA under KCl, LiCl, or PDS conditions. **e, f** CD (**e**) and FRET (**f**) melting measurements of PQS-675-WT and PQS-675-Mut RNA under KCl, LiCl, or PDS conditions. The melting temperature is shown in Supplementary Table 3. **g** Gel mobility shift assay of PQS-675-WT and PQS-675-Mut RNA under KCl or LiCl conditions. **h** BLI sensorgrams show the binding affinity of PDS with PQS-675-WT (top panel) and PQS-675-Mut (bottom panel) RNA under KCl condition. _KD_ binding affinity, _Ka_ binding rate, _Kd_ dissociation rate. **i** Representative confocal fluorescence microphotographs of FAM-labeled RNA (green) and BG4 (red) treated with PDS or ASO (targeting PQS-675 RG4 site) in H1299 cells. The nuclei were stained with DAPI (blue). The colocalized FAM/BG4 foci are indicated by white boxes. Scale bars: 5 μm. **j** Quantification of FAM/BG4 foci number after FAM-labeled RNA transfection in **h**. For each condition, 20–30 cells were counted, and the standard error of the mean was calculated from a set of three replicates. Data are shown as mean ± SEM, $n = 3$. Two-tailed Student's $t$ test. Source data are provided as a source data file.

fold into RG4 structure (Fig. 1c). In contrast, the intensity of all the characteristic peaks was decreased when mutated the RG4 sites in PQS-675 or dissolved into LiCl conditions (Fig. 1c and Supplementary Fig 1d). Despite this decrease, there was strong fluorescence of ThT for PQS-675-Mut (Fig. 1c), consistent with previous observations that ThT staining may have non-specific effects rather than RG4[34,35]. Therefore, the capability of PQS-675 for RG4 formation was further confirmed by the fluorescence resonance energy transfer (FRET) assay, as evidenced by the increased FRET level of this PQS, but not its mutant, in KCl (Fig. 1d). Moreover, both CD- and FRET-melting assays were used to evaluate the thermostability of PQS-675. These analyses consistently revealed that the Tm of PQS-675 was increased in the presence of KCl as compared with LiCl (Fig. 1e, f and Supplementary Table 3). In addition, recent evidence suggests that the flanking sequences of RG4 region may influence RG4 formation and stability[36,37]. To test this possibility, we generated the construct containing four additional nucleotides on either side of the PQS-675 RG4-forming sequence (PQS-675FL, Supplementary Fig 1d). PQS-675FL showed typical positive and negative molar ellipticity peaks at 264 nm and 238 nm, albeit its spectrum intensity and Tm were lower than PQS-675 (Supplementary Fig. 1g), supporting the capability of this PQS for RG4 formation in a longer context, as well as the potential effect of flanking sequences on its formation and stability.

We next performed a gel mobility shift assay using FAM-labeled PQS-675 RNAs to monitor structure compaction. The wild type (WT) and G4-mutant (Mut) RNAs were folded in the presence of LiCl or KCl, and the structures were analyzed by native PAGE. Under KCl condition, the WT RNAs, but not the mutants, displayed a retarded migrating band that was slower than ssRNA band, and this retarded band was greatly diminished under LiCl condition (Fig. 1g), further supporting the RG4 formation of PQS-675.

Of note, multiple ligands can specifically bind and stabilize RG4 configurations, with pyridostatin (PDS) being one powerful example[25]. Accordingly, PDS treatment amplified CD emission and FRET level of PQS-675 and increased its Tm in the presence of KCl, however, these effects were attenuated in LiCl (Figs. 1d–f, Supplementary Fig. 1h and Supplementary Table 3). In line with the specificity of PDS on RG4 stabilization, these effects were not observed in PQS-675 mutants (Fig. 1d, f and Supplementary Table 3). Furthermore, bio-layer interferometry (BLI) assay verified the high-affinity binding of PDS with PQS-675-WT RNA, and this interaction was diminished by the mutation of RG4 sequences (Fig. 1h), indicating that PDS can directly interact with PQS-675 and stabilize its RG4 structure.

Finally, we detected RG4 formation in living cells by immunofluorescence assay[38]. FAM-labeled WT or G4-mutant PQS-675 RNAs were transfected into H1299 cells, and then RG4s were visualized by the G4-specific antibody BG4[39]. Substantial colocalization of WT RNAs and RG4s was detected, whereas this

colocalization was nearly disappeared in cells transfected with G4-mutant RNAs (Fig. 1i, j), indicating the ability of RG4 formation for PQS-675 in living cells. As an RG4 stabilizer, PDS enhanced this colocalization in cells transfected with the WT RNAs (Fig. 1i, j). Conversely, antisense oligonucleotides (ASOs) that were partly complementary to PQS-675 sequences reduced this colocalization (Fig. 1i, j), in line with the notion that ASOs could unwind RG4 structures[40].

Taken together, these results suggest that PQS-675 (hereafter, referred to as GQS-675) in _Tmprss2_ has the potential to form RG4 structure in vitro and in vivo.

**RG4 inhibits _Tmprss2_ translation.** Based on the above results, TMPRSS2, particularly its GQS-675, was used to exemplify the potential role of RG4 in SARS-CoV-2 pathogenesis. Initially, we examined the possible effect of RG4 in TMPRSS2 expression in vivo. The SARS-CoV-2 susceptible H1299 cells that endogenously express _Ace2_ and _Tmprss2_ were treated with different RG4 stabilizers, and then the protein and mRNA levels of _Tmprss2_ were measured. All RG4 stabilizers, including PDS, carboxypyridostatin (cPDS), and TMPyP4, reduced TMPRSS2 protein levels, but had no effect on its mRNA levels (Fig. 2a and Supplementary Fig. 2a), suggesting a post-transcriptional inhibitory role of RG4 in TMPRSS2 expression. However, it is noteworthy that TMPyP4 can also destabilize RG4s[41], thus its effects on GQS-675 should be carefully evaluated with caution in the future.

We then clarified whether the _Tmprss2_ repression triggered by RG4 stabilizers correlates with GQS-675. We generated pcDNA3.1 plasmids containing the full-length ORF of _Tmprss2_ (hG4WT) or G4-mutant _Tmprss2_ (hG4mut1), in which Gs in the middle two G-tracts of GQS-675 were substituted with adenines (As) (Fig. 2b). This mutation was designed to eliminate the RG4 formation with synonymous substitution. Transfection of both plasmids into H1299 cells led to a comparable increase in _Tmprss2_ mRNAs (Fig. 2c and Supplementary Fig. 2b). However, the protein level of TMPRSS2 was higher in hG4mut1 cells than hG4WT cells (Fig. 2c and Supplementary Fig. 2b), suggesting an inhibitory role of GQS-675 in post-transcriptional regulation. Notably, RG4 stabilizers, including PDS, cPDS, and TMPyP4, diminished the increase of TMPRSS2 expression by hG4WT plasmids at the post-transcriptional level (Fig. 2d and Supplementary Fig. 2c), but not by hG4mut1 plasmids (Fig. 2e and Supplementary Fig. 2d), indicating that the G/A mutations can greatly abolish the inhibition of RG4 on TMPRSS2 expression. These observations were recapitulated in human bronchial epithelial cells (HBE) that also endogenously express _Tmprss2_ and _Ace2_ (Supplementary Fig. 2e–h). To exclude the potential interference of endogenous _Tmprss2_, we performed similar experiments in hACE2-293T cells that normally do not express _Tmprss2_, and obtained similar results (Fig. 2f–h and Supplementary Fig. 2i–k). To clarify if these differences between hG4WT and hG4mut1 are due to potential codon

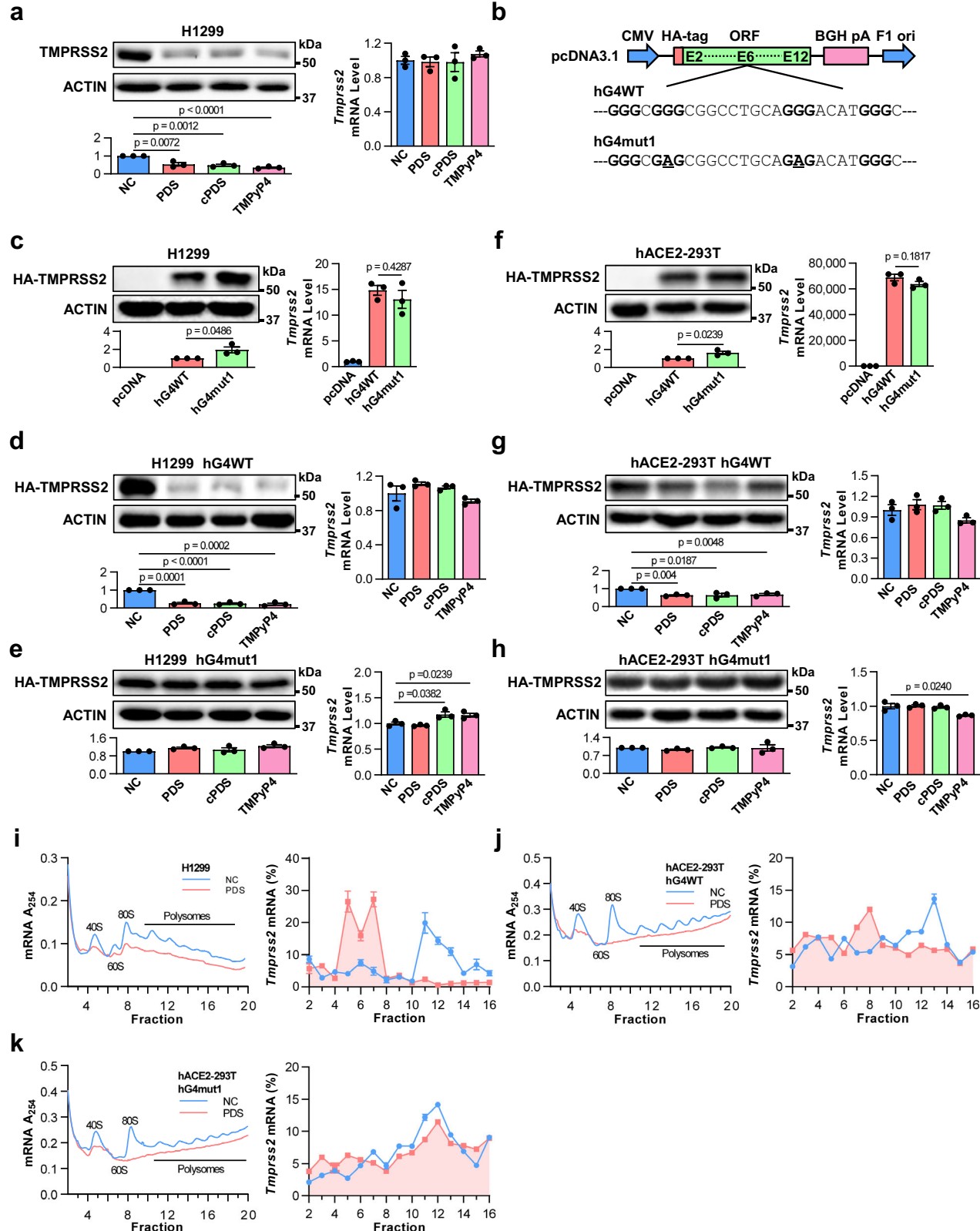

usage bias, we further constructed the hG4mut2 and hG4mut3 plasmids, in which Gs in the middle two G-tracts of GQS-675 were substituted with thymines (Ts) and cytosines (Cs), respectively (Supplementary Fig. 2l). Similar to hG4mut1, both hG4mut2 and hG4mut3 plasmids resulted in an increase in TMPRSS2 protein levels compared with the hG4WT plasmids, and this increase was not affected by RG4 stabilizers (Supplementary Fig. 2m), arguing against

the potential contribution of codon usage bias on hG4mut-induced TMPRSS2 expression. Together, these data demonstrate that the formation of RG4 in GQS-675 can efficiently inhibit TMPRSS2 expression at the post-transcriptional level.

RG4s in ORFs often behave as translation regulators[40,42–44]. To test this possibility, we analyzed the ribosomal distribution profiles of endogenous cytosolic *Tmprss2* mRNAs in H1299 cells

**Fig. 2 RG4 inhibits TMPRSS2 translation. a** Levels of endogenous TMPRSS2 protein (left panel), and mRNA (right panel) in H1299 cells treated with PDS (10 μM), cPDS (10 μM), or TMPyP4 (10 μM). ImageJ quantification of the TMPRSS2/ACTIN ratio is shown (bottom panel). **b** The full-length ORF region of human *Tmprss2* or its RG4-mutant sequence was inserted into the MCS of pcDNA3.1 vector. The bold letters represent the G-tracts and the underlined letters represent mutations. **c** Levels of HA-TMPRSS2 protein (left panel), and mRNA (right panel) in H1299 cells transfected with hG4WT or hG4mut1 plasmids. ImageJ quantification of the HA-TMPRSS2/ACTIN ratio is shown (bottom panel). **d, e** Levels of HA-TMPRSS2 protein (left panel), and mRNA (right panel) in H1299 cells transfected with hG4WT (**d**) or hG4mut1 (**e**) plasmids, and stimulated with PDS (10 μM), cPDS (10 μM), or TMPyP4 (10 μM). ImageJ quantification of the HA-TMPRSS2/ACTIN ratio is shown (bottom panel). **f** Levels of HA-TMPRSS2 protein (left panel), and mRNA (right panel) in hACE2-293T cells transfected with hG4WT or hG4mut1 plasmids. ImageJ quantification of the HA-TMPRSS2/ACTIN ratio is shown (bottom panel). **g, h** Levels of HA-TMPRSS2 protein (left panel), and mRNA (right panel) in hACE2-293T cells transfected with hG4WT (**g**) or hG4mut1 (**h**) plasmids, and stimulated with PDS (10 μM), cPDS (10 μM), or TMPyP4 (10 μM). ImageJ quantification of the HA-TMPRSS2/ACTIN ratio is shown (bottom panel). **i** Polysome shift analysis of endogenous *Tmprss2* mRNA in H1299 cells. Representative polysome gradient profile shows the position of polysomes, monosomes (80 S), small (40 S), and large (60 S) subunits (left panel). *Tmprss2* expression in control and PDS-treated cells was calculated as the percentage of mRNA in each fraction compared with total mRNA of all fractions (right panel). **j, k** Polysome shift analysis of exogenous *Tmprss2* mRNA in hG4WT (**j**) and hG4mut1 (**k**) transfected hACE2-293T cells. A representative polysome gradient profile was shown on the left. *Tmprss2* expression in each fraction from control and PDS-treated cells was shown on the right. Data are shown as mean ± SEM, *n* = 3. Two-tailed Student's *t* test. Source data are provided as a source data file.

treated with PDS. Polysome shift analysis showed that PDS treatment resulted in a shift of *Tmprss2* mRNA peak towards the light polysome fraction (Fig. 2i), indicating an inefficient *Tmprss2* translation caused by RG4 formation. Furthermore, PDS treatment also led to a decrease in exogenous *Tmprss2* translation in G4WT-transfected hACE2-293T cells (Fig. 2j) Interestingly, PDS appeared to suppress mRNA amounts associated with both heavy and light polysome fractions in both H1299 and hACE2-293T cells, indicative of decreased global translation (Fig. 2i, j), which is consistent with previous observations in other cell lines[45]. To further clarify that the translational inhibition of PDS on *Tmprss2* is associated with RG4s within *Tmprss2* instead of global translation attenuation, we analyzed the translation profile of exogenous *Tmprss2* in hG4mut1-transfected hACE2-293T cells. Although PDS led to a global translation block in these cells, it did not obviously alter the translation profile of *Tmprss2* mutant mRNAs (Fig. 2k), supporting the importance of RG4s in PDS-mediated *Tmprss2* suppression. As a specificity control, actin mRNAs without PQSs were actively translated in hACE2-293T cells, regardless of PDS treatment (Supplementary Fig. 2n). Taken together, these results suggest that RG4 can impede *Tmprss2* translation in living cells.

**RG4 in *Tmprss2* inhibits SARS-CoV-2 entry in cells.** Given that RG4s are presented in both SARS-CoV-2 and host factors, as well as a single RG4 is sufficient to reduce TMPRSS2 expression, we postulated that RG4 may regulate SARS-CoV-2 infection. To test this notion, a recently established pseudovirus system, in which vesicular stomatitis virus (VSV) was pseudotyped with SARS-CoV-2 S glycoprotein (i.e., SARS-CoV-2-S-luc), was used to monitor the viral entry step[7,46]. The pseudoviruses then infected human lung cells and expressed a fluorescence protein Renilla for quantification (Fig. 3a). The luciferase activity showed that all three RG4 stabilizers significantly reduced pseudovirus entry efficiency in both H1299 and HBE cells (Fig. 3b and Supplementary Fig. 3a). Intriguingly, the inhibition efficiency of PDS was comparable to that of camostat mesylate (Fig. 3b), a known anti-COVID-19 drug. Of note, the antiviral activity was not owing to cellular cytotoxicity, since these RG4 stabilizers had no effect on cell viability (Supplementary Fig. 3b).

We further clarified the influence of GQS-675 in virus entry by using the aforementioned hG4WT and hG4mut1 plasmids. In line with TMPRSS2 protein level (Fig. 2c, f), the pseudovirus entry efficiency was higher in hG4mut1 cells than hG4WT cells (Fig. 3c, d). Moreover, PDS treatment reduced pseudovirus entry in hG4WT cells, and this inhibition was abolished in hG4mut1 cells

(Fig. 3e, f), indicating a potent inhibitory impact of GQS-675 on virus entry.

Together, these results suggest that stabilization of RG4 can effectively reduce SARS-CoV-2 infection.

**Stabilizing RG4 hinders SARS-CoV-2 infection in vivo.** We next investigated the potential effect of RG4 on SARS-CoV-2 infection in mouse models. The murine *Tmprss2* is well-conserved with the human homolog and harbors multiple PQSs in ORF (Fig. 4a and Supplementary Table 4). The top three PQSs in mouse *Tmprss2* were used for RG4 characterization (Fig. 4a, b). Among them, PQS-1370 and PQS-1585 were shown to adopt RG4 structure, as evidenced by in silico and biochemical methods including CD and fluorescence emission spectrum measurements (Fig. 4b and Supplementary Fig. 4a–c). Moreover, CD-melting revealed that the Tm of PQS-1370 was higher in KCl than that in LiCl (Fig. 4c), indicative of a thermostable RG4 structure.

Subsequently, we assessed the effect of RG4 on murine TMPRSS2 expression. RG4 stabilizers were capable to reduce endogenous TMPRSS2 expression in murine Lewis lung carcinoma cells (LLC) (Fig. 4d). By generating the murine *Tmprss2* (mG4WT) and its G4-mutant (mG4mut) plasmids (Fig. 4e), we further showed the level of exogenous TMPRSS2 was higher in mG4mut cells than mG4WT cells (Fig. 4f, g). Moreover, RG4 stabilizers diminished the induction of TMPRSS2 by mG4WT plasmids, but not by mG4mut plasmids, further supporting the inhibitory role of RG4 in murine TMPRSS2 expression (Fig. 4h, i). In addition, the human *Tmprss2* and its RG4 mutants were heterogeneously expressed in LLC cells, respectively, and the impact of PDS on their expression was assessed. All three of the RG4-mutant plasmids (hG4mut1/2/3) led to an increase in human TMPRSS2 expression compared with the hG4WT plasmids (Supplementary Fig. 4d). More importantly, the heterogeneous expression of human *Tmprss2*, but not its RG4 mutants, was repressed by RG4 stabilizers (Supplementary Fig. 4e). These results suggest the conserved role of RG4 on TMPRSS2 expression in mice, making it possible to detect SARS-CoV-2 infection in vivo.

To this end, we first analyzed the toxicity of PDS in mice. There were no mice that died during 9 days of continued injection of low dose (6 mg/kg) or high dose (30 mg/kg) PDS, and no abnormal behaviors or bodyweight loss were observed in the low-dose group (Supplementary Fig. 5a). Moreover, mice treated with PDS showed no obvious clinically relevant changes in histological or hematological parameters (Fig. 5b and Supplementary Fig. 5b), suggesting low toxicity of PDS in vivo.

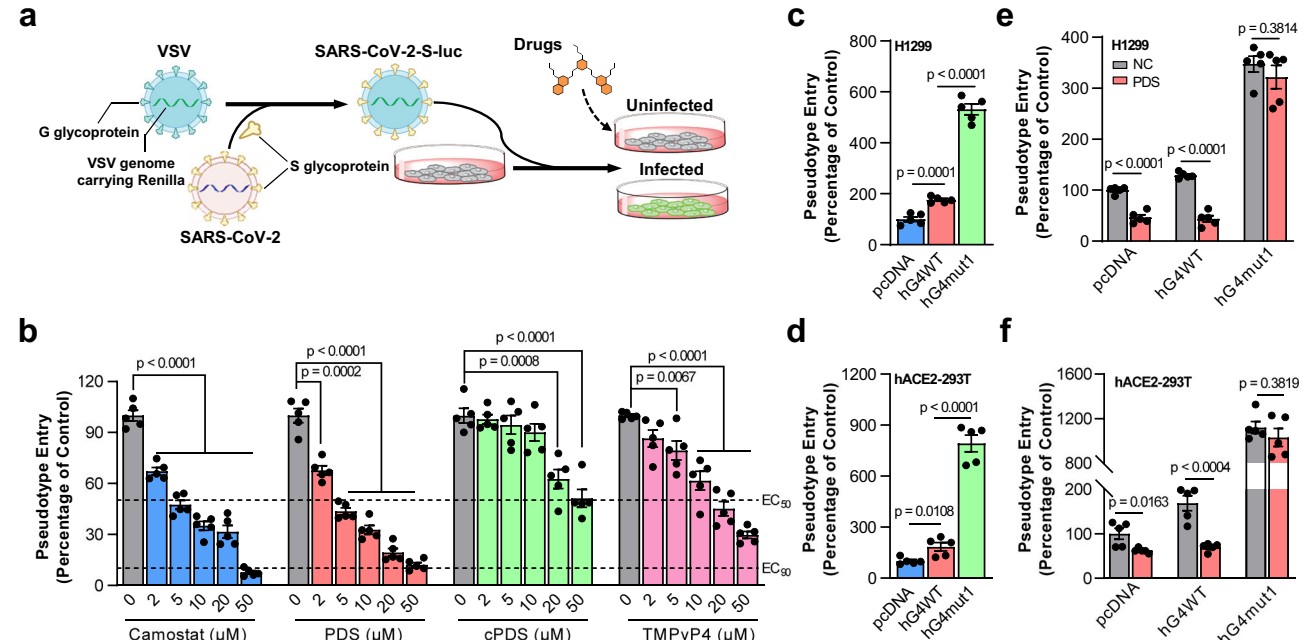

**Fig. 3 RG4 inhibits SARS-CoV-2 pseudoviruses entry in vitro. a** Schematic of the SARS-CoV-2 pseudoviruses entry system. The VSV viruses were pseudotyped with SARS-CoV-2 S glycoprotein (i.e., SARS-CoV-2-S-luc), and cells infected with SARS-CoV-2-S-luc for 48 h were then examined for luciferase activity. **b** SARS-CoV-2 pseudoviruses entry efficiency with camostat mesylate, PDS, cPDS, or TMPyP4 treatment. H1299 cells were pre-incubated with the indicated concentrations of compounds, and subsequently inoculated with VSV-SARS-2-S-luc. 48 h post-inoculation, pseudoviruses entry was analyzed by determining luciferase activity in cell lysates (normalization against untreated cells). **c, d** Entry efficiency of SARS-CoV-2 pseudoviruses in H1299 (**c**) and hACE2-293T (**d**) cells transfected with hG4WT or hG4mut1 plasmids. **e, f** Entry efficiency of SARS-CoV-2 pseudoviruses in H1299 (**e**) and hACE2-293T (**f**) cells transfected with hG4WT or hG4mut1 plasmids and stimulated with PDS (10 μM). Data are shown as mean ± SEM, $n = 5$. Two-tailed Student's $t$ test. Source data are provided as a source data file.

It has been reported that laboratory mice cannot be infected by SARS-CoV-2 due to the virus's inability to use the mouse ACE2 for S glycoprotein binding[47]. In this circumstance, C57BL/6 J mice were heterogeneously expressed of hACE2 via adeno-associated virus (AAV) as described previously[48], and challenged with VSV-SARS-2-S-luc. In brief, mice were initially inoculated with $5 \times 10^{11}$ genomic copies (GCs) of AAV9-hACE2, and then infected with $2.5 \times 10^8$ relative light units (RLUs) of VSV-SARS-2-S-luc pseudovirus 7 days after. From −1 day post infection (DPI), mice were injected with PDS daily and killed at 8 DPI (Fig. 5a). Compared with saline-treated mice, PDS-treated mice had a significant reduction in luminescent intensity (Fig. 5c, d) and *Renilla* expression (Fig. 5e), suggesting decreased entry of pseudovirus. These results indicate a protective role of RG4 stabilizers against SARS-CoV-2 infection in vivo. In line with the inhibition of pseudovirus entry, PDS administration led to a decrease in TMPRSS2 protein levels in lungs, as determined by immunohistochemical (IHC) (Fig. 5b), enzyme-linked immunosorbent assay (ELISA) (Fig. 5f), and western blot (Fig. 5g, h). Since the fluorescence signals were also observed in the abdomen, we determined the levels of hepatic TMPRSS2 protein and found a similar reduction in PDS-treated mice (Supplementary Fig. 5d, e). Collectively, these results strongly suggest that RG4 stabilization in vivo may effectively hinder SARS-CoV-2 infection.

**TMPRSS2 is induced in the lungs of patients with COVID-19.** Finally, we linked the relevance of our experimental findings to a clinical condition. Given the significance of RG4 on TMPRSS2 regulation and virus infection, we hypothesized that TMPRSS2 upregulation in susceptible cells at the site of infection may contribute to SARS-CoV-2 pathogenesis. To test this idea, we

performed IHC staining to evaluate the expression of TMPRSS2 in human lung tissues from two healthy individuals and two patients who succumbed to COVID-19. TMPRSS2 was rare and limited to the bronchus and alveoli of healthy lung tissues, whereas it was increased in patients with COVID-19 (Supplementary Fig. 6a). In line with this, high levels of TMPRSS2 expression were observed in the lungs of nine autopsy patients with COVID-19[49], whereas its expression was barely detected in the lungs of individuals without COVID-19 from the HPA database (Human Protein Altas, https://www.proteinatlas.org/) (Supplementary Fig. 6b). Together, these results suggest that increased expression of host factors might contribute to SARS-CoV-2 infection and pathogenesis.

## Discussion

In this study, we demonstrate the role of RG4 in SARS-CoV-2 infection. Both SARS-CoV-2 genome and host factors harbor multiple PQSs that can fold into canonical or non-canonical RG4 structures. As exemplified by *Tmprss2*, we identify the biological, physiological, and pathological functions of RG4 in gene regulation and SARS-CoV-2 infection (Fig. 5i). These findings reveal a post-transcriptional regulatory mechanism underlying SARS-CoV-2 pathogenesis and thus offer a promising strategy for the prevention and treatment of COVID-19.

RG4 has vital roles in various biological and physiological processes and participates in the pathogenesis of multiple virus diseases[25]. The viral G4s are usually located in regulatory regions of the genome and have been implicated in the control of key viral processes. Here, we predicted numerous PQSs in SARS-CoV-2 genome and further verified a non-canonical RG4 structure within NSP10, suggesting a role of RG4 in SARS-CoV-2 pathogenesis. Interestingly, all PQSs within

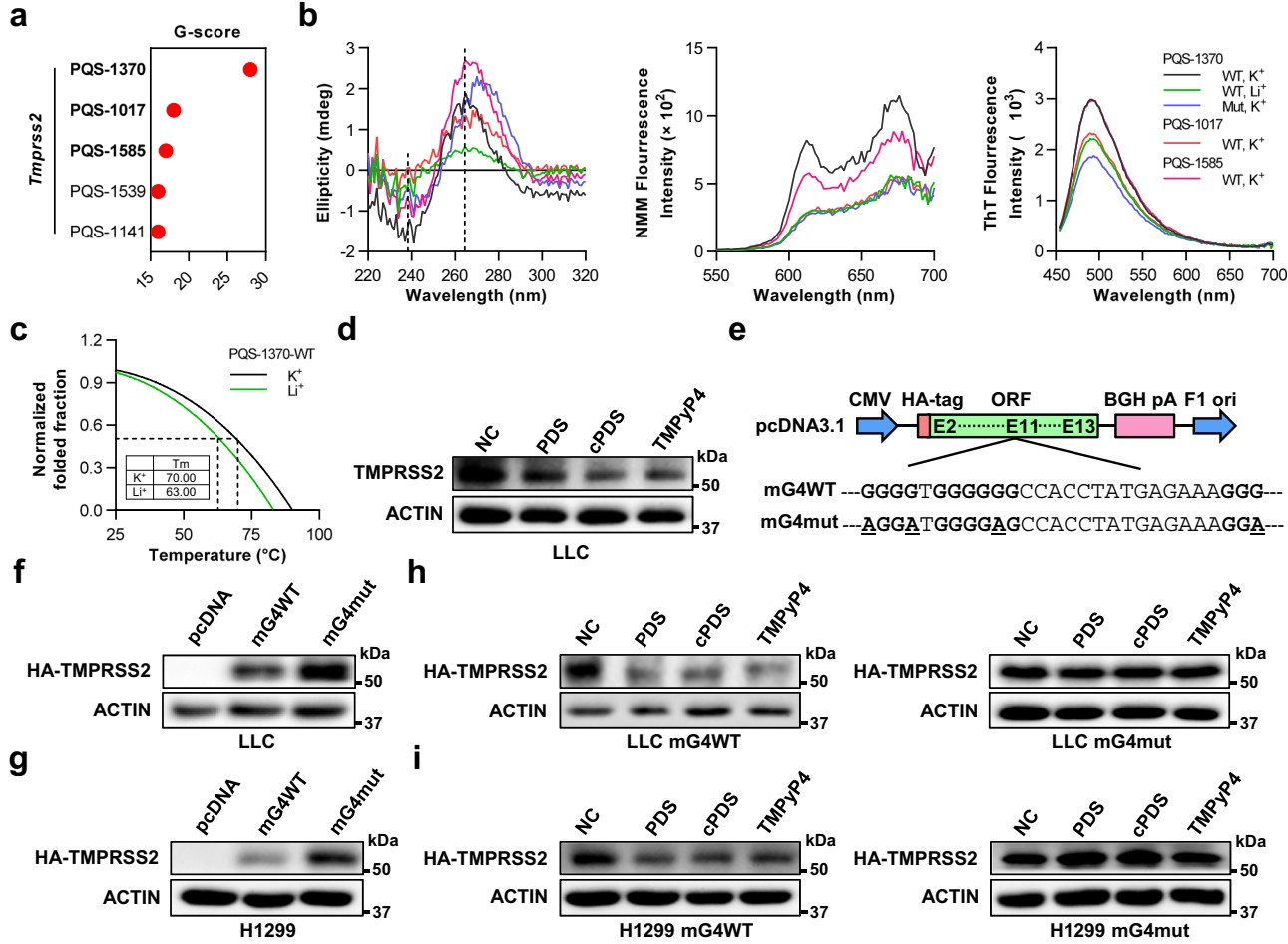

**Fig. 4 RG4 in murine *Tmprss2* mRNA inhibits its protein expression. a** RG4 potential of mouse *Tmprss2* mRNA was predicted by QGRS-Mapper. **b** CD spectrum (left panel), NMM (middle panel), and ThT (right panel) fluorescence emission spectra of PQS-1017, PQS-1370, and PQS-1585 RNA under KCl or LiCl conditions. **c** CD-melting measurements of PQS-1370-WT RNA under KCl or LiCl conditions. **d** Levels of endogenous TMPRSS2 protein in LLC cells treated with PDS (10 μM), cPDS (10 μM), or TMPyP4 (10 μM). **e** The full-length ORF region of murine *Tmprss2* or its RG4-mutant sequence was inserted into the MCS of pcDNA3.1 vector. The bold letters represent the G-tracts and the underlined letters represent mutations. **f, g** Levels of HA-TMPRSS2 protein in LLC (**f**) and H1299 (**g**) cells transfected with mG4WT or mG4mut plasmids. **h, i** Levels of HA-TMPRSS2 protein in LLC (**h**) and H1299 (**i**) cells transfected with mG4WT (left panel) or mG4mut (right panel) plasmids, and stimulated with PDS (10 μM), cPDS (10 μM), or TMPyP4 (10 μM). Source data are provided as a source data file.

SARS-CoV-2 genome seem to adopt non-canonical two-quartet structures that are more metastable than three-quartet RG4s. The metastable characteristic could enable the virus to rapidly modulate its expression and activity in response to different intracellular factors or physiological conditions in host cells. During the preparation of this manuscript, three papers also reported the existence of RG4 in SARS-CoV-2 genome[26–28]. Consistent with our results, Ji et al. identified a metastable RG4 structure formed by PQS-13385 in NSP10[27]. PQS-28903, which was neglected in our study, has been reported to form RG4 structure and inhibit Nucleocapsid expression[26]. Collectively, our present study, together with other recent works, suggests a potential involvement of RG4 in SARS-CoV-2 regulation.

Furthermore, we provided several lines of evidence, for the first time, to demonstrate the presence and functional significance of RG4 in host factors of SARS-CoV-2. The coronavirus receptor ACE2 and the activator TMPRSS2 have been recognized as key molecules in SARS-CoV-2 host cell infection[7]. Bioinformatic analysis predicted multiple PQSs within *Ace2* and *Tmprss2*, and some of them were confirmed to adopt canonical or non-canonical RG4 structures by a combination of biochemical and biophysical approaches. Specifically, GQS-675 in the ORF of

*Tmprss2* can form a thermodynamic stabilized three-quartet RG4 structure and significantly repress *Tmprss2* translation, thereby blocking TMPRSS2-primed host cell entry of SARS-CoV-2 in vivo. Of note, additional host factors have recently been found to be relevant for SARS-CoV-2 entry, including Axl[50], furin[51], cathepsins-L[52], and neuropilin-1[53,54]. Interestingly, these newly characterized host factors also have the potential to adopt RG4 structures (Supplementary Fig. 7), indicating a prevalent regulatory role of RG4 in SARS-CoV-2 host factors.

More importantly, we explored the clinical implications of RG4 in SARS-CoV-2 pathology in vitro and in vivo by using G4-specific stabilizers that represent emerging drugs[26]. Strikingly, the G4-specific compound PDS can prevent pseudovirus entry into host cells, comparable to that of camostat mesylate, a drug being used for COVID-19[7]. Moreover, the 9-day administration of PDS in mice resulted in a reduction in pseudovirus infection in vivo, accompanied by a decrease in TMPRSS2 protein level. These results define an inhibitory effect of RG4-specific stabilizers on SARS-CoV-2 in mouse models. Thus, our present and previous studies[24] strongly suggest the effectiveness and low toxicity of PDS in vivo and provide new insights into novel strategies for the prevention and treatment of human diseases, including

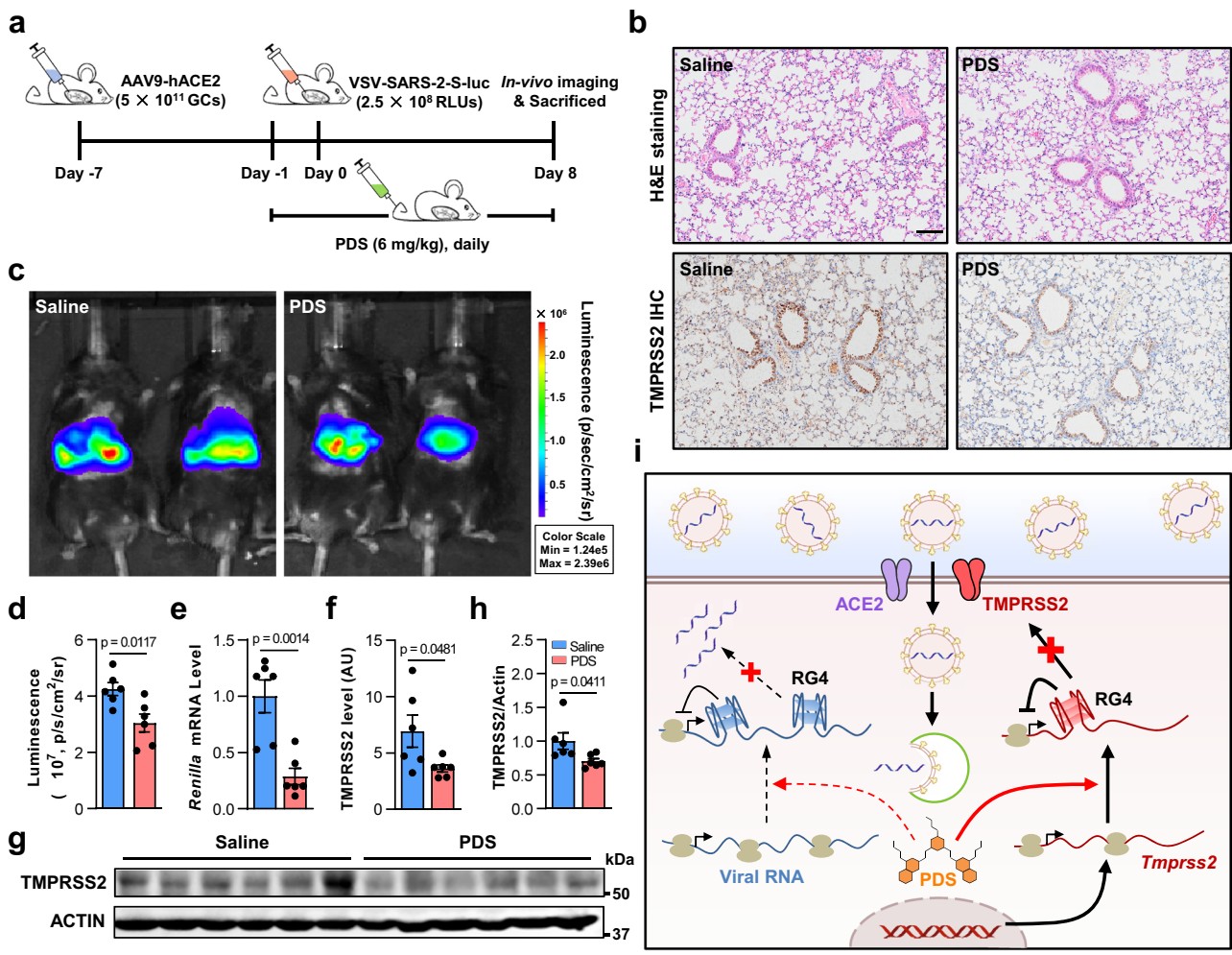

**Fig. 5 RG4 hinders the efficacy of SARS-CoV-2 infection in vivo. a** Schematic of in vivo experimental plans. 6–8-week-old C57BL/6 J mice were transduced intrathoracic with an AAV coding for human ACE2 (AAV9-hACE2) and infected with VSV-SARS-2-S-luc after 7 days. From one day before the infection, mice were infused with PDS (6 mg/kg body weight) or saline via caudal vein daily. Following measurements were performed on day 8 post infection (n = 6). **b** Representative H&E staining (top panel) and IHC staining for TMPRSS2 (bottom panel) in paraffin lung sections. Scale bars: 100 μm. **c** Representative photos of the VSV-SARS-2-S-luc-infected mice. The relative levels of bioluminescence are shown in pseudocolors, with blue and red representing the weakest and strongest photon fluxes, respectively. **d** Photon emission from each mouse was quantified as radiance (p/s/cm²/sr) post-imaging, using region of interest analysis in living image software. **e** Levels of Renilla mRNA in the VSV-SARS-2-S-luc-infected mouse lungs. **f–h** Levels of TMPRSS2 protein in the VSV-SARS-2-S-luc-infected mouse lungs analyzed by ELISA (**f**) or western blot (**g**). ImageJ quantification of the TMPRSS2/ACTIN ratio in (**g**) is shown (**h**). **i** A schematic model of RG4 regulation in SARS-CoV-2 infection. Data are shown as mean ± SEM, n = 6. Two-tailed Student's t test. Source data are provided as a source data file.

COVID-19. Of note, in principle, PDS can bind numerous G4s in the host and virus without target specificity, thus new strategies targeting TMPRSS2 RG4 are needed to clarify this regulation in SARS-CoV-2 infection. Unfortunately, there are still no clinical trials on agents based on RG4 mechanism yet. Thus, it is of importance for future investigation to develop gene-specific RG4-targeting agents/strategies. In addition, although we provided some evidence on the low toxicity of PDS in mice, its druggability and clinical utility should be carefully evaluated in the future. Finally, given that both the SARS-CoV-2 and its host factors contain numerous putative RG4s, it is highly anticipated that G4-targeting agents are also able to inhibit the infection of some SARS-CoV-2 variants, if not all.

Despite the functional significance of *Tmprss2* in COVID-19 pathogenesis, the molecular mechanisms underlying its endogenous regulation remain largely unknown. Consistent with the inhibitory role of RG4 in *Tmprss2* translation and SARS-CoV-2 infection presented here, our results indicated that the protein level of TMPRSS2 was increased in the lungs of COVID-19 patients. Interestingly, a recent RNA-seq analysis suggested a decrease in *Tmprss2* mRNAs post SARS-CoV-2 infection[55], consistent with the observation that the expression of host genes is often extremely inhibited in SARS-CoV-2-infected cells[56]. RG4 primarily regulates gene expression via diverse post-transcriptional mechanisms[10], including pre-mRNA splicing[20], mRNA localization[19], and translation[18,21]. Among them, trans-lational control might provide quicker adaptive responses to environmental cues than other steps involved in gene regulation[57]. Here we did find that a single RG4 in *Tmprss2* ORF can lead to a dramatic decrease in TMPRSS2 proteins, implying potent inhibition on translation. Under these circumstances, it is possible that TMPRSS2, as well as other host factors, could be regulated at both transcriptional and post-transcriptional levels, whereas the latter may conduct the determinant effect on their expression and function. Given the importance of RG4 for *Tmprss2* translation and SARS-CoV-2 infection, it is of interest to

identify the regulator(s) modulating *Tmprss2* RG4. On the one hand, a number of known RG4 helicases, such as DHX9, DHX36, DDX3X, DDX5, and DDX21, were elevated in SARS-CoV-2-infected host cells[58,59], suggesting that these helicases may play a role in the regulation of RG4s within the host and/or virus genomes, including *Tmprss2*. On the other hand, certain components of SARS-CoV-2 genome, such as NSP3[60] and NSP13[27] that could unwind RG4 structures, may participate in modulating RG4 dynamics in both viruses and host genomes during SARS-CoV-2 pathogenesis. Anyway, the precise alternation of TMPRSS2 mRNA and proteins in response to SARS-CoV-2 infection awaits further investigation.

Nevertheless, several limitations remain to be addressed in the future. For instance, the pseudovirus system can only simulate the process of viral entry. The authentic SARS-CoV-2 assay, or even primary lung cells from COVID-19 patients if possible, is still required to verify the systematic effect of RG4s on SARS-CoV-2 lifecycle. In addition, we observed both pulmonic and hepatic targeting of VSV-SARS-2-S-luc in AAV9-hACE2-mice, probably due to the pulmonic and hepatic tropism of AAV9, as well as the abundance of hepatic TMPRSS2. In this regard, the use of AAV systems specific to the lung could further clarify our findings. Last but not least, the clinical link between RG4 and SARS-CoV-2 pathogenesis presented in this study is limited by a small number of COVID-19 patients. Determining the mRNA and protein level of cell host factors in a large cohort is of importance to elucidate their dynamic regulation mechanisms, including RG4.

In summary, this study reveals a previously unknown pathophysiological mechanism underlying SARS-CoV-2 lifecycle and highlights the potential clinical translation for RG4 to antiviral intervention.

## Methods

**Clinical specimens**. All human studies were conducted according to the principles of the Declaration of Helsinki, and approved by the Institutional Review Board and Biomedical Ethics Committee of West China Hospital of Sichuan University (WCH/SCU) (2020, no. 126). We have obtained informed consent from all participants.

COVID-19 lung tissues were obtained from the Division of Respiratory and Critical Care Medicine at WCH/SCU. Two patients were selected for pathologic studies based on a respiratory tract sample positive for SARS-CoV-2 in a Quantitative Real-time PCR (QRT-PCR) assay, as tested by a designated diagnostic laboratory (Patient1: female, 75 years old; patient2: female, 71 years old). Paraffin-embedded sections were sectioned by the Department of Pathology at WCH/SCU. Sections of healthy people's lungs without underlying chronic airway diseases were obtained from Shanghai Outdo Biotech (China).

**Animals**. In all, 6–8-week-old male C57BL/6 J mice were purchased from Nanjing Biomedical Research Institute (China). Mice were fed a standard laboratory chow diet, water ad libitum under a 12 h light–dark cycle (lights on from 8 a.m. to 8 p.m.) at constant temperature (22°C). All mouse-related experiments were conducted according to the protocols approved by the Institutional Animal Care and Use Committee of WCH/SCU (20220125002).

**Cell culture**. H1299 (CRL-5803) and HEK-293T (CRL-3216) were purchased from the American type culture collection (ATCC). HBE (4201PAT-CCTCC00691) and LLC (1101MOU-PUMC000673) were obtained from the National infrastructure of cell line resource (NICLR). All cell lines were cultured in Dulbecco's Modified Eagle Medium (10569044, Gibco, USA) supplemented with 10% fetal bovine serum (30044184, Gibco, USA) and 1% Penicillin–Streptomycin (15140122, Gibco, USA), and grown in a 37°C incubator with 5% $CO_2$.

For RG4 stabilization, cells were treated with 10 μM of PDS (HY-15176A, MCE, China), cPDS (SML1176, Sigma-Aldrich, USA), or TMPyP4 (613560, Sigma-Aldrich, USA) for 48 h before RNA or protein extraction. For cell viability and pseudotype entry tests, cells were treated with camostat mesylate (HY-13512, MCE, China), PDS, cPDS, or TMPyP4 for indicated times and doses.

**Oligonucleotides and antibodies**. DNA and RNA oligonucleotides (Supplementary Tables 5 and 6) were purchased from TSINGKE (China) and GenePharma (China) respectively. Antibodies used in this study were listed as follows: anti-TMPRSS2 antibody (A9126, Abclonal, China), anti-HA-tag antibody (3724S, CST, USA), anti-β-actin antibody (60008-1-IG, Proteintech, USA), and BG4 (Ab00174-24.1, Absolute Antibody, UK).

**AAV infection and SARS-CoV-2 pseudovirus infection**. AAV9-encoding hACE2 (hACE2-AAV9) and SARS-CoV-2 spike glycoprotein pseudotyping vesicular stomatitis virus (SARS-CoV-2-S-luc) were purchased from Delivectory Biosciences (China). These SARS-CoV-2-S-luc pseudotyped viral particles consist of a lentiviral core with Renilla element, and the SARS-CoV-2 S glycoprotein on its envelope. Therefore, cells infected with pseudoviruses can be examined via luciferase activity.

Mice were anesthetized using a mixture of ketamine (50 mg/kg) and xylazine (5 mg/kg) by intraperitoneal injection. The chest was shaved and disinfected, and then animals were inoculated via an intrathoracic (IT) route with $5 \times 10^{11}$ GCs of AAV9-hACE2. 7 days after, hACE2-mice were challenged with an IT injection of $2.5 \times 10^8$ RLUs of VSV-SARS-2-S-luc pseudovirus. For PDS administration, from −1 DPI, mice were injected with PDS (6 mg/kg body weight) intravenously once a day, and performed following detection at 8 DPI.

**In vivo bioluminescence imaging**. Mice were injected intraperitoneally with D-Luciferin (P1041, Promega, USA) in phosphate-buffered saline (PBS) at a 150 mg/kg body weight dose, before subjection to general anesthesia using 30% isoflurane inhalation in the induction chamber. 5 to 10 min after, mice were placed in the IVIS Spectrum imaging chamber (PerkinElmer, USA), and images were captured while anesthetized. The radiance was determined via the region of interest analysis using Living Image Analyze 12.0 (PerkinElmer, USA).

**Hematoxylin and eosin (H&E) staining**. Fresh tissues were collected and immersed in 10% neutral-buffered formalin for fixation, followed by paraffin embedment. Fixed tissues were sectioned into 4 μm-thick and stained with H&E for morphological analyses.

**IHC staining**. The pulmonic and hepatic sections were incubated with anti-TMPRSS2 antibodies, followed by binding with horseradish peroxidase (HRP)-conjugated secondary antibodies, and detected with DAB Kit (ZLI-9017, Zsbio, China) and counterstained with hematoxylin.

**Serum biochemistry**. Blood sampling was performed from mice by cardiac puncture. Serum was obtained by centrifuging clotted blood samples at $3000 \times g$ for 15 min at 4 °C. Serum biochemical indices were determined using Cobas8000 automatic analyzer (Roche, USA).

**Plasmids, transfection, and lentivirus transduction**. The *Tmprss2* cDNA was PCR amplified and cloned into pcDNA3.1 vector. For G4mut plasmid, Gs in the middle two G-tracts of GQS-675 in *Tmprss2* were substituted with A/T/Cs to eliminate the RG4 formation with synonymous substitution.

Cells were seeded into 6- or 12-well plates (ThermoFisher, USA) and transfected with indicated plasmids using Attractene (1051563, QIAGEN, USA) according to the manufacturer's protocol.

The hACE2-293T was established by lentivirus transduction. The human *Ace2* was PCR amplified and cloned into pLVX-IRES-BSD vector. Transduction with lentiviral particles was performed using polybrene (TR-1003, Sigma-Aldrich, USA), and positive cells were selected with blasticidin (203350, Sigma-Aldrich, USA).

**SARS-CoV-2 spike-mediated pseudovirus entry assay**. For in cellulo pseudovirus transduction, host cells were cultured in 96-well plates and inoculated with VSV-SARS-2-S-luc pseudovirus (MOI = 1) at 60–80% confluency. For experiments involving RG4 stabilizers and camostat mesylate, cells were treated with the respective chemical 5 h before transduction. For experiments involving plasmids, cells were transfected with indicated plasmids 24 h before transduction. After transduction of 16 h, the culture medium was replaced with a fresh medium. Transduction efficiency was then quantified 48 h post transduction by measuring the activity of *Renilla* luciferase in cell lysates using the ONE-Glo™ Luciferase Assay (E6120, Promega, USA) according to the manufacturer's instructions.

**Quantification of cell viability**. Cell viability of H1299 cells following treatment with camostat mesylate and RG4 stabilizers was determined by 3-[4,5-dimethyl-thiazol-2-yl]-5-[3-carboxymethoxyphenyl]-2-[4-sulfophenyl]-2H-tetrazolium (MTS) analysis[61]. In brief, 10,000 cells were seeded in a 96-well plate and incubated for 24 h in the absence or presence of different concentrations of camostat mesylate, PDS, cPDS, or TMPyP4. Then cell viability was determined using the CellTiter 96 AQueous Nonradioactive Cell Proliferation Assay (G5421, Promega, USA) according to the manufacturer's instructions.

**Immunofluorescence**. Immunofluorescence was performed as previously described with minor modifications[24]. In brief, H1299 cells were transfected with FAM-labeled RNAs (Supplementary Table 5) combined with or without 100 nM 20-mer ASO by HiPerFect transfection reagent (301705, Qiagen, USA). 24 h later, cells were fixed, permeabilized, and incubated with BG4 antibody, followed by binding with anti-goat Alexa 647-conjugated antibodies (ab150131, Abcam, UK) and 4′,6-diamidino-2-phenylindole counterstaining. Fluorescence was detected by FV3000

confocal laser scanning microscope (Olympus, Japan). For RG4 stabilization, cells were treated with 10 μM of PDS for 18 h before fixation.

**RNA extraction and QRT-PCR.** Total RNAs were extracted by Tri-Reagent (TR118, MRC, USA) according to the manufacturer's instructions. cDNA was synthesized using M-MLV Reverse Transcriptase (28025021, ThermoFisher, USA). QRT-PCR was performed using QuantiNova SYBR Green PCR Kit (208052, QIAGEN, USA) with specific primers (Supplementary Table 6). Expression of mRNA was normalized to actin.

**Western blot analysis.** Total proteins were extracted using RIPA lysis buffer (89901, ThermoFisher, USA) with protease inhibitors (11873580001, Roche, USA) and quantified by the Bradford assay (500-0205, Bio-Rad, USA). Equivalent denatured samples were separated by sodium dodecyl sulfate–polyacrylamide gel electrophoresis and transferred to the PVDF membrane (A10122278, GE, USA). The membrane was blocked with 5% non-fat milk, incubated with primary antibodies and followed by HRP-conjugated secondary antibodies, and then developed using Pierce™ ECL Western Blotting Substrate (34076, ThermoFisher, USA).

**ELISA.** The protein level of TMPRSS2 in the lungs of mice who suffered from SARS-CoV-2-S-luc pseudovirus was determined by ELISA using a kit (DL-TMPRSS2-Mu, DLDEVELOP, China) according to the manufacturer's instructions. The OD value at 450 nm was measured with Hybrid Multi-Mode Reader (Bioteck, USA).

**Polysome profiling.** Polysome profiling was performed as previously described with minor modifications[62,63]. In brief, ~$1.5 \times 10^7$ cells were incubated with 100 μg/ml cycloheximide (CHX, S7418, Selleck, China) in growth media for 10 min at 37 °C and 5% $CO_2$, washed twice with precooling PBS (containing 100 μg/ml CHX), harvested and centrifuged at $500 \times g$ for 5 min at 4 °C. Cells were then resuspended in 425 μl hypotonic buffer (containing 5 mM Tris-HCl pH 7.5, 2.5 mM $MgCl_2$, 1.5 mM KCl, 100 μg/ml CHX, 1 μl of 1 M DTT, RNase inhibitors and protease inhibitor cocktail) and vortexed for 5 sec. Then 25 μl of 10% Triton X-100 and 25 μl of 10% sodium deoxycholate were added to cells followed by vortexing for 5 s. Centrifuge lysates at $16,000 \times g$ for 7 min at 4 °C and transfer supernatant to a new pre-chilled 1.5 ml tube. The same $OD_{260}$ amount of lysates from each sample was loaded onto a 5–50% sucrose gradient and centrifuged at $222,228 \times g$ for 2 h at 4 °C using SW40Ti rotor with Beckman L8-M Ultracentrifuge (Beckman, USA). Samples were analyzed on the Piston Gradient Fractionator (Biocomp Instruments, Canada) and Fraction collector (Gilson Inc, Canada). Data were analyzed using the Piston Gradient Fractionator Data View software. Polysome fractions were collected at 0.6 ml/fraction and RNAs were extracted by Tri-Reagent. For PDS stimulation, cells were treated with 10 μM PDS for 24 h. For plasmids transfection, cells were transfected with indicated plasmids for 48 h.

**CD spectrum and melting temperature measurements.** In all, 5 μM RNAs (Supplementary Table 5) were dissolved in 10 mM Tris-HCl (pH 7.5) containing 150 mM KCl or LiCl, heated at 95 °C for 5 min, and then gradually cooled down to 4 °C. The CD spectra were measured by Chirascan-Plus CD Spectrometer (Applied Photophysics, UK) according to our previous study[24]. For CD-melting experiments, the ellipticity of annealed RNAs at 264 nm was monitored on continuous heating at 1 °C/min between 25 °C and 90 °C.

**Fluorescence emission spectrum measurements.** In all, 2 μM RNAs (Supplementary Table 5) were folded in 150 mM KCl or LiCl as described above. In all, 2 μM NMM (NMM580, Frontier Science, USA) or ThT (HY-D0218, MCE, China) were then added. The sample was excited at 393 nm for NMM and 425 nm for ThT. The emission spectrum was collected from 500 nm to 700 nm for NMM, and 450–700 nm for ThT, respectively. Fluorescence spectroscopy was performed using Hybrid Multi-Mode Reader (Bioteck, USA) with a reaction volume of 100 μl.

**FRET spectrum and melting temperature measurements.** 5′-FAM and 3′-TAMRA dual labeled RNAs (Supplementary Table 5) at a final concentration of 200 nM were folded in 150 mM KCl or LiCl as described above. For molecules, 1 μM PDS was added to the reaction buffer before annealing. The FRET spectra were measured with an excitation wavelength of 483 nm and a detection wavelength varied from 510 to 650 nm using Hybrid Multi-Mode Reader (Bioteck, USA). For the FRET-melting experiments, fluorescence readings with excitation at 483 nm and detection at 578 nm were performed across a temperature range from 25 °C to 95 °C, with a 0.02 °C per second temperature gradient. The fluorescence melting curves were determined using a QuantStudio (TM) 7 Flex system (ThermoFisher, USA).

**Gel mobility shift assay.** 5′-FAM-labeled RNAs were folded as described above and electrophoresed on a 15% native polyacrylamide gel at 4 °C. The gel was imaged using Phosphorimager Typhoon FLA9500 (GE, USA).

**BLI assay.** The interaction between PQS-675 and PDS was detected in black 96-well microplates (Greiner, Germany) at 25 °C using ForteBio Octet RED 96e (Sartorius, Germany) and Super Streptavidin (SSA) biosensors (Sartorius, Germany). PBST (DEPC-treated PBS, 0.01% Tween-20) buffer was used as assay buffer to minimize non-specific interactions for BLI studies. SSA sensors were hydrated for 10 min, and the baseline optical interference was recorded. Biotin-labeled RNAs (200 nM) (Supplementary Table 5) were folded as described above and loaded onto the SSA sensor for 2 min, then washed in PBST buffer at a shaking speed of 1000 rpm for 5 min. After that, binding interaction with different concentrations of PDS (0.5 μM, 1 μM, 2 μM, 4 μM, 8 μM) was performed, which consisted of baseline (120 s), association (540 s), and dissociation (600 s). A reference sensor without RNA served as the background control. Curves were fit to a 1:1 interaction model, and the Kd value was calculated using Octet data analysis studio software (Sartorius, Germany).

**Statistical analysis.** All data represented at least three independent experiments and were shown as mean ± SEM. Statistical analysis was performed using Graphpad Prism 8 software. Two-tailed unpaired Student's $t$ test was used to determine the difference between two independent groups unless otherwise indicated, and $p < 0.05$ was considered significant.

**Reporting summary.** Further information on research design is available in the Nature Research Reporting Summary linked to this article.

## Data availability

The data that support the findings of this study are available from the corresponding author upon reasonable request. Source data are provided with this paper.

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

## Acknowledgements

We thank the staff from the Institute of Clinical Pathology (Li Li, Fei Chen, and Chunjuan Bao) and Core Research Facilities (Yu Ding, Li Fu, Cong Li, Yanjing Zhang, Yinchan Wang, Shuaishuai Yu, and Sisi Wu) of West China Hospital for their continuous support. We also thank Chunyu Cao from China Three Gorges University for his technical support of mouse experiments. This work was supported by National Natural Science Foundation of China (92157205 and 81970561 to X.F.; 82172986 to Y.T.; 82173182 to D.L.; 82000547 to G.L.); Ministry of Science and Technology (2018ZX09201018-005 to X.F.); the 1.3.5 Project for Disciplines of Excellence, West China Hospital, Sichuan University (ZYJC18049 to X.F.); National Clinical Research Center for Geriatrics, West China Hospital, Sichuan University (Z20191005 to X.F.); the Fellowship of China Postdoctoral Science Foundation (2020TQ0215, 2021M690112 to G.L.) and Sichuan University Postdoctoral Interdisciplinary Innovation Fund to G.L.

## Author contributions

X.F. and G.L. conceptualized and designed the experiments. X.F. and Y.T. supervised the study. G.L. performed most of the experiments with W.D., X.S., and Q.T. as assistants. G.C. and Y.Y. provided critical mouse materials. W.C., Y. W., L.J., and D.L. provided clinical specimens and data. X.F., G.L., and W.D. wrote the manuscript. All authors read and approved the final version of the manuscript.

## Competing interests

The authors declare no competing interests.
