## [Peer Review File · Nature Communications]

REVIEWER COMMENTS

Reviewer #1 (Remarks to the Author):

In this study, Liu et. al reported the role of RNA G-quadruplex (RG4) in SARS-CoV-2 infection. Combining bioinformatics, fluorescence, CD and native gel assays, they demonstrate the presence of RG4s in both SARS-CoV-2 genome and host factors. They have shown the canonical 3-quartet RG4 within Tmprss2 can inhibit Tmprss2 translation and prevent SARS-CoV-2 entry. They also demonstrated that G4-specific stabilizers can attenuate SARS-CoV-2 infection in pseudovirus cell systems and mouse models, suggesting Tmprss2 rG4 maybe a potential target for COVID-19 prevention and treatment. Overall, the results of the manuscript are of potential interest to the research of pathophysiological mechanism and potential drug development of SARS-CoV-2. However, there are some additional experiments that must be performed to strengthen the results and claims they made in this manuscript. Also, several issues should be further clarified and discussed by the authors. Below please find my comments that should help to improve the flow and quality of the manuscript, and I suggest the authors should fully address them.

Major comments

1. From the fluorescence and CD data in Figure 1c, the G4 structure in PQS-675 do not seem to be strong (low CD/fluorescence intensity, small difference between K⁺ and Li⁺), so I suggest the authors perform CD/UV melting, or FRET melting, to further verify the G4 formation and its thermostability. For example, what's its T_m under physiological relevant K⁺ condition (~150mM)?
2. To make sure PQS-675 can bind and be stabilized by G4 stabilizer, I suggest the authors test the binding of PQS-675 to RG4 stabilizers (e.g. PDS, cPDS, TMPyP4) in buffer to support the cell data in Figure 2. The increase in thermostability (T_m increase), if any, can be measured by CD/UV melting, or FRET melting.
3. Figure 2. RG4 inhibits Tmpress2 translation. In this test, the authors generated pcDNA3.1 plasmids containing the full-length ORF of Tmprss2 (G4WT) or G4-mutant Tmprss2 (G4mut), in which guanylic acids in the middle two G-tracts of PQS-675 were substituted with adenylic acids (Fig. 2b). This mutation was designed to eliminate the RG4 formation with synonymous substitution. How to make sure the difference in protein expression between the wildtype and mutant is owing to RG4 effect instead of potential codon usage bias? The author may generate other synonymous mutation to support it experimentally. Otherwise, the author needs to discuss the potential caveats of this experiment in the discussion.
4. Figure 2: The authors used the general G4 ligands to inhibit TMPRSS2 translation. Besides targeting rG4 in the TMPRSS2 mRNA, those ligands can also bind to other DNA/RNA G4s in the host and virus. The author needs to address this issue on how rG4 in TMPRSS2 can be a good target for SARS-COV-2 under these DNA/RNA G4 background. Also, does the ligand treatment affect the global translation, subsets of transcripts that contains rG4, or just TMPRSS2 translation? This is not clear from the data presented at the moment, and the author need to further demonstrate and discuss clearly.
5. Figure 5 and Supplementary Figure 5. The authors only show 2 healthy and 2 patient lung tissue samples, which is not strong enough sample size to support the findings here. More samples are required with proper statistical test performed.
6. Figure 5. The authors showed TMPRSS2 is induced in lung of patient with COVID-19, but there is no

experimental evidence provided by the author that G4 (or addition of G4 stabilizer) can downregulate the expression of TMPRSS2 in patient cells, nor a potential G4 regulator (e.g. protein or helicase) being identified, making this a correlation. More experiments and discussion are needed to substantiate the proposed model.

Minor comments

1. In this work, the author characterized the RG4 in SARS-CoV-2 genome at the beginning. However, the presence of RG4s in SARS-CoV-2 genome has been reported before (Reference 25, 26, 27). While the author has cited them, it will be important to point out the difference and similarity in the prediction and in vitro experimental validation among the findings in this work and the previous reports to help the readers.
2. From the native gel in Figure 1d, the PQS-675-Mut only shows the ssRNA band but no RG4 band. However, in Figure 1c, PQS-675-Mut shows enhanced fluorescence with G4 ligand NMM/ThT, and specifically, it has relatively strong fluorescence with ThT. The author needs to address this.
3. Figure 2a and 2d: Did the author use the same concentration of RG4 stabilizer for Figure 2a and Figure 2d? Why does the reduced TMPRSS2 protein levels in Figure 2d is much more significant than Figure 2a?
4. Figure 2c: Why there is no TMPRSS2 expression in the pcDNA control (lane 1)? There should be some endogenous TMPRSS2 in H1299 cell.
5. Figure 3: The authors should give more details on the design of pseudovirus luciferase system, and it is better to include a scheme.
6. Figure 4: The authors should give more details on the design of the VSV-SARS-2-S-luc-infected mouse models.
7. Page 17. 'Interestingly, these newly characterized host factors also have the potential to adopt RG4 structures (data not shown), indicating a prevalent regulatory role of RG4 in SARS-CoV-2 host factors'. I suggest the authors show these data (e.g. QGRS score, conservation, etc) or cite reference here.
8. In some places in the manuscript, some typos were spotted. For example, 'SRAS-CoV-2' in Page 14 should be SARS-CoV-2.

Reviewer #2 (Remarks to the Author):

This manuscript by Liu et al. assesses the contribution of RNA G-quadruplexes (RG4s) in modulating SARS-CoV-2 infection. The authors mainly focus on characterising the role of a single RG4 motif found within the ORF of the Tmprss2 protease known to regulate SARS-CoV-2 entry and show that stabilisation of this RG4 motif, using small molecules, inhibit Tmprss2 translation and SARS-CoV-2 infection both in vitro and in vivo. This manuscript is timely and may be of great interest for identifying new approaches to tackle SARS-CoV-2 infection. Nevertheless, some crucial experiments and controls are needed to fully support the conclusions of the manuscript. Here are some comments/suggestions:

Main points:

- (1) The biophysical characterisation of viral and host RG4s is lacking important experiments and controls

to fully support RG4 formation and stabilisation by the small molecules. The authors should report the thermal stability of each studied RG4 motif and demonstrate the potassium dependency of RG4 stability. CD spectra are not sufficient to support RG4 formation (for example the CD spectra of PQS-13385 and PQS-24268 are identical while only the former is proposed to fold into RG4). The authors should also study the stability of RG4s (especially PQS-675) in a longer context as it is known that flanking sequences impact greatly both folding and stability. Finally, it is crucial to demonstrate that the three small molecules (PDS, cPDS and TMPyP4) are able to bind to and stabilise PQS-675 but not its mutated version. It is noteworthy that TMPyP4 have been reported for destabilising RG4s and enhance translation (see Morris et al. NAR 2012, 40, 4137-4145 for example).

(2) The detection of RG4 in living cells by immunofluorescence assay is lacking of quantification. The authors should report the quantification of FAM/BG4 foci number after FAM-labelled RNAs transfection. The authors should also demonstrate that an antisense oligonucleotide to the RG4 and treatment with a small molecule stabiliser decrease and increase the colocalisation of the signal respectively.

(3) The impact of small molecules on Tmprss2 translation is somehow inconsistent and the authors need to clarify some of the results. The efficiency of the small molecules seems indeed to be cell- and/or target-specific. For example, PDS has little to no effect on inhibiting the expression of endogenous Tmprss2 in H1299 cells (Fig. 2a), seems to be highly efficient in inhibiting an ectopically expressed Tmprss2 in the same cell line (Fig. 2d) and has no effect on the translation of the same construct in hACE2-293T cells (Fig. 2g). Such discrepancies are observed for cPDS and TMPyP4. It is unclear what is the exact protocol the authors have followed and the number of replicates that have been performed, but the authors should report additional data to support that their findings are robust and reproducible. The authors should report all WB replicates together with the quantification of band intensity in a supplementary figure for the reader to be able to appreciate the variability associated with these experiments.

(4) The authors assessed the impact of PDS on the distribution of ribosomes within the Tmprss2 mRNA in H1299 and hACE2-293T cells. Again, the observations in both cell lines are not consistent. While PDS shift the mRNA peak toward monosomes in both cell lines, the authors observed an accumulation of 40S and 60S monosomes in H1299 cells but an accumulation of 80S monosomes in hACE2-293T cells. This point should be clarified. The authors report polysome gradient profiles (A254) only for untreated cells but should report similar profiles after PDS treatment to demonstrate that PDS is not a general inhibitor of translation. Finally, the authors should demonstrate that this shift of Tmprss2 mRNA towards the monosome fraction is RG4-dependent by studying the G4mut-transfected hACE2-293T cells. This latter control is crucial to appreciate the contribution of RG4 to Tmprss2 translation.

(5) In Fig. 3c, it is surprising that PDS has a similar inhibitory effect in cells transfected with the empty or the G4WT vector. This observation suggests that overexpression of Tmprss2 does not stimulate virus entry. In order to clarify this point, the authors should perform the same experiment using G4WT- and G4mut-transfected hACE2 293T cells that do not endogenously express Tmprss2.

(6) A major caveat with the in vivo experiments is that the PQS-675 RG4 found within the human Tmprss2 is not present in the corresponding murine mRNA. The authors then postulate that all their

findings related to the human Tmprss2 mRNA are effective for the murine Tmprss2 mRNA. This hypothesis must carefully be validated by reproducing key experiments with the murine mRNA. Experiments, such as the ones described in Fig. 2 and 3, should be reproduced with vectors encoding the murine Tmprss2 comprising WT and mutated RG4 motifs. Similarly, the authors should assess the impact of PDS on the translation of the WT and mutated human Tmprss2 when heterogeneously expressed in mice. These experiments are crucial to support the role of RG4s in modulating SARS-CoV-2 entry in vivo.

(7) Differences in Tmprss2 protein levels (saline vs PDS) on the Western blot reported in Fig. 4i are difficult to appreciate. The authors should quantify the intensity of the bands and perform a statistical test to assess the relevance of their result. Because this result is central to this manuscript, the author should consider quantifying the difference in protein level using a more quantitative technique such as ELISA.

(8) Because RG4s are found within the virus genome (S protein) and within the human ACE2 receptor, the authors need to assess whether PDS affect the transcription or translation of both viral and host factors to demonstrate that the observed decrease in SARS-CoV-2 infection in vivo is solely due to the inhibition of Tmprss2 translation. Without these data, the current observations are purely descriptive and not causative.

(9) The authors wrote in the discussion section: “Thus, our present and previous studies strongly suggest the effectiveness and safety of PDS in vivo”. This statement is surprising as the acute toxicity of PDS and its ability to trigger genetic instability is well documented. To support their claim the authors should report the impact of PDS treatment on mice fitness and lifespan. The clinical and hematological parameters reported in Supplementary Figure 4e. are not very informative.

Minor points:

(10) The authors may want to consider changing the title of their manuscript to reflect that their work mainly focuses on Tmprss2 mRNA translation inhibition rather than targeting the SARS-CoV-2 viral genome. As it stands the title may be misleading.

(11) When describing sequence mutants, the authors should refer to the nucleobases (i.e. guanine and adenine) rather than the corresponding monophosphate (guanylic and adenylic acids).

(12) In the introduction, the authors report that the UK variant is about “70% more transmittable”, but this point hasn’t been formally demonstrated in the population. This should be clarified.

(13) The authors should avoid the use of words such as “extraordinary”, “remarkably” and other superlatives.

Reviewer #3 (Remarks to the Author):

The manuscript by Liu et al. reports the role of RG4 in SARS-CoV-2 infection. Initially, the authors predicted G4-forming sequences in SARS-CoV-2 genome and major host factors for viral entry by bioinformatic analysis, and experimentally confirmed the existence of RG4 within SARS-CoV-2 genome in vitro. Next, the existence and function of RG4 in host factors regulating SARS-CoV-2 infection was exemplified by a canonical RG4 within Tmprss2. Combining several chemical, biochemical, and molecular methods, the authors provided several lines of clear and solid evidence to demonstrate that this RG4 can be formed and inhibit Tmprss2 translation in vitro and in vivo. Moreover, they demonstrated that this RG4 in Tmprss2 can inhibit SARS-CoV-2 entry in cells and reduce SARS-CoV-2 infection in mice by using the pseudovirus system. In addition, the authors found a reduction of Tmprss2 protein in the lung of COVID-19 patients.

This work provides the first evidence to demonstrate the role of RG4 in SARS-CoV-2 infection in vitro and in vivo by using the pseudovirus system. The study is well done, and the data are novel and solid. The written is clear and the flow is excellent. Overall, this timely and highly significant study could provide new ideas against the ongoing COVID-19 pandemic. Nevertheless, there are few concerns that the authors need to address for the manuscript to warrant publication

Major points:

1. The data presented in Fig. 3c clearly showed that PDS treatment impairs SARS-CoV-2 infection in H1299 cells transfected with G4WT plasmids, but not with G4mut plasmids. This data is critical to support that the inhibition of RG4 formation within Tmprss2 can suppress SARS-CoV-2 infection. In this regard, it is recommended to perform the similar experiment in an additional cell line, such as hACE2-293T, to exclude the potential cell-specific effect and thus strengthen the conclusion.
2. The authors used IHC analysis and found that Tmprss2 protein is reduced in the lung of two COVID-19 patients compared with two healthy controls (Figure 5). The sample size was small. It is understandable since the lung tissue of COVID-19 patients is very difficult to obtain and Tmprss2 mRNA level is impracticable to be determined in paraffin-embedded tissues. However, the authors should, at least, clearly state and discuss these limitations in the text, if no more lung tissues of COVID-19 patients can be obtained.
3. The authors showed that PDS treatment can decrease exogenous Tmprss2 translation in G4WT-transfected hACE2-293T cells by using polysome shift analysis (Figure 2j). To further confirm that this effect depends on the G4 sequence, the authors should perform the similar experiment in hACE2-293T cells transfected G4mut plasmids.

Minor point:

1. The quantification of some representative results, such as Fig 1e, Fig 2 (western blots), and Fig 4i, should also be shown.
2. The authors showed that the PQS-675 can repress Tmprss2 translation and function. The sequence of PQS-675 appears to be evolutionarily conserved, which is summarized in Figure 1a. However, it is better to also present the conservation of this PQS-675 by sequence alignment among different species, which can be included in the Supplementary Data.
3. By using polysome shift analysis, the authors found that PDS treatment resulted in a decrease in endogenous Tmprss2 mRNA in H1299 cells (Figure 2i), as well as in exogenous Tmprss2 mRNA in hACE2-293T cells (Figure 2j). However, the effect of PDS on the percentage of Tmprss2 mRNA in these two cell

lines seems to be different. What is the explanation?

4. As mentioned by the authors, some newly characterized host factors also have the potential to adopt RG4 structures (data not shown). Given that the broad effect of PDS on RG4 structures, it is better to appropriately show this prediction data in Supplemental Data.

5. This study showed that the RG4 stabilizer PDS can reduce SARS-CoV-2 infection in mouse models. The authors are encouraged to briefly discuss the potential of PDS, as well as other RG4 stabilizers, in translational medicine and its clinical implication. Additionally, G4-forming sequences within SARS-CoV-2 genome have been predicted by this study and other recent reports. Currently, some SARS-CoV-2 variants are emerging globally. It is of interest to discuss the possible implication of RG4 on SARS-CoV-2 variants, if applicable.

6. As mentioned by the authors, the existence of RG4 within SARS-CoV-2 genome has been reported recently (Ref No. 25-27 in the main text). In this regard, the authors are encouraged to describe the differences between this study and previous literatures, which may be helpful to clarify the novelty and importance of their work, as well as to help the readers for understanding the topic.

7. Numerous RNA binding proteins (RBPs) have been shown to stabilize or unwind RG4 structures. The identification of potential RBPs responsible for the dynamics of RG4s within SARS-CoV-2 genome and host factors, particularly Tmprss2, requires intense investigation and may be beyond the scope of this study, however the authors are encouraged to appropriately introduce and discuss this interesting topic, which could throw new insights on further investigation.

8. The abbreviation of "PQS", suggesting instead of "GQS".

9. It is not clear about G4 when it was first presented, clarify.

10. The last sentence of introduction need to be modified.

Dear Editors and Reviewers,

We are very grateful to your encouraging and insightful comments/suggestions regarding our original submission. In response to these comments, we have made a number of modifications to our manuscript and performed additional suggested experiments. Below we detail these modifications with the specific comments from the reviewers (in *italic*), followed by our response (in **blue**). We have highlighted all changes underlined in the revised manuscript. We hope the reviewers will find this manuscript improved following these changes and more suitable for publication in ***Nature Communications***.

The point-by-point responses to all of the reviewers' comments are listed below:

Reviewer 1

In this study, Liu et. al reported the role of RNA G-quadruplex (RG4) in SARS-CoV-2 infection. Combining bioinformatics, fluorescence, CD and native gel assays, they demonstrate the presence of RG4s in both SARS-CoV-2 genome and host factors. They have shown the canonical 3-quartet RG4 within Tmprss2 can inhibit Tmprss2 translation and prevent SARS-CoV-2 entry. They also demonstrated that G4-specific stabilizers can attenuate SARS-CoV-2 infection in pseudovirus cell systems and mouse models, suggesting Tmprss2 rG4 maybe a potential target for COVID-19 prevention and treatment.

Overall, the results of the manuscript are of potential interest to the research of pathophysiological mechanism and potential drug development of SARS-CoV-2. However, there are some additional experiments that must be performed to strengthen the results and claims they made in this manuscript. Also, several issues should be further clarified and discussed by the authors. Below please find my comments that should help to improve the flow and quality of the manuscript, and I suggest the authors should fully address them.

Response: We appreciate the reviewer for the favorable and insightful comments.

Major comments

1. From the fluorescence and CD data in Figure 1c, the G4 structure in PQS-675 do not seem to be strong (low CD/fluorescence intensity, small difference between K⁺ and Li⁺), so I suggest the authors perform CD/UV melting, or FRET melting, to further verify the G4 formation and its thermostability. For example, what's its T_m under physiological relevant K⁺ condition (~150mM)?

Response: Thank the reviewer for this critical comment. Inspired by the thoughtful comments from both reviewers (**Reviewer 2, Point 1**), we have verified the formation and thermostability of the RG4 structure in PQS-675 by using FRET- and CD-melting assays (**Figure below**). The FRET level of PQS-675 was increased in the presence of KCl compared to LiCl, supporting the capability for RG4 formation. Moreover, both FRET- and CD-melting assays consistently showed that the melting temperature (T_m) of PQS-675 was increased in the physiologically relevant levels of KCl (150 mM) compared to LiCl. Together, these data strengthen that PQS-675 can form an energetically favorable RG4 structure. These results are now described in **Fig. 1d, 1e, 1f, Supplementary Fig. 1h, and Supplementary table 3**.

Supplementary Table 3

Group	T _m (CD-melting)	T _m (FRET-melting)
PQS-675-WT		
K ⁺	70.00	74.00
K ⁺ , PDS	74.00	77.48
Li ⁺	64.00	63.27
Li ⁺ , PDS	64.00	65.71
PQS-675-MUT		
K ⁺	N/A	38.33
K ⁺ , PDS	N/A	38.03

2. To make sure PQS-675 can bind and be stabilized by G4 stabilizer, I suggest the authors test the binding of PQS-675 to RG4 stabilizers (e.g. PDS, cPDS, TMPyP4) in buffer to support the cell data in Figure 2. The increase in thermostability (T_m increase), if any, can be measured by CD/UV melting, or FRET melting.

Response: Thank the reviewer for this crucial suggestion, which is also raised by **Reviewer 2 (Point 1)**. Given that PDS is one of most popular RG4 stabilizer and predominantly used in our study both *in vitro* and *in vivo*, we have determined its effects on PQS-675 RG4 formation and thermostability by using CD- and FRET-melting assays. PDS treatment amplified CD emission and FRET level of PQS-675, and increased its T_m in KCl condition (**see also Reviewer 1, Point 1**). Furthermore, we have used Bio-Layer Interferometry (BLI) assay to verify the high RG4-binding affinity of PDS with PQS-675-WT RNA, but not PQS-675-mutant RNA (**Figure below**). These new data indicate that PDS can directly interact with PQS-675 and stabilize its RG4 structure. These results are now described in **Fig. 1d, 1e, 1f, 1h, Supplementary Fig. 1h, and Supplementary table 3**.

Fig.1h

3. Figure 2. RG4 inhibits Tmprss2 translation. In this test, the authors generated pcDNA3.1 plasmids containing the full-length ORF of Tmprss2 (G4WT) or G4-mutant Tmprss2 (G4mut), in which guanylic acids in the middle two G-tracts of PQS-675 were substituted with adenylic acids (Fig. 2b). This mutation was designed to eliminate the RG4 formation with synonymous substitution. How to make sure the difference in protein

expression between the wildtype and mutant is owing to RG4 effect instead of potential codon usage bias? The author may generate other synonymous mutation to support it experimentally. Otherwise, the author needs to discuss the potential caveats of this experiment in the discussion.

Response: Thank the reviewer for this thoughtful comment. We fully agree with the reviewer that it is of great necessity to consider the potential codon usage bias on *Tmprss2* translation. By following the suggestion, we have constructed two additional mutations of TMPRSS2, namely hG4mut2 and hG4mut3, in which guanines (Gs) in the middle two G-tracts of PQS-675 were substituted with thymines (Ts) and cytosines (Cs), respectively. Similar as hG4mut1 (known as G4mut in the original manuscript), both hG4mut2 and hG4mut3 plasmids resulted in an increase in TMPRSS2 protein levels compared to the hG4WT plasmids, and this increase was not affected by RG4 stabilizers (**Figure below**), arguing against the potential contribution of codon usage bias on hG4mut-induced TMPRSS2 expression. These results are now described in **Supplementary Fig. 2l**, and **2m**.

Supplementary Fig. 2l

Supplementary Fig. 2m

4. Figure 2: The authors used the general G4 ligands to inhibit TMPRSS2 translation. Besides targeting rG4 in the TMPRSS2 mRNA, those ligands can also bind to other DNA/RNA G4s in the host and virus. The author needs to address this issue on how rG4 in TMPRSS2 can be a good target for SARS-COV-2 under these DNA/RNA G4 background. Also, does the ligand treatment affect the global translation, subsets of transcripts that

contains rG4, or just TMPRSS2 translation? This is not clear from the data presented at the moment, and the author need to further demonstrate and discuss clearly.

Response: We appreciate the reviewer for this insightful comment, which is also raised by **Reviewer 2 (Point 4)**. Given that thousands of putative RG4 regions have been identified in human transcriptome (*Science*, 2016, PMID: 27708011; *Nat Methods*, 2016, PMID: 27571552), and that PDS is a RG4 stabilizer without target specificity (*Nat Chem*, 2010, PMID: 21107376), we totally agree with the reviewer that PDS could bind other RG4s in the host and virus. As mentioned in the manuscript, *Ace2* and *Tmprss2*, two most common cellular entry determinants for SARS-CoV-2, were predicted to have multiple PQSs with considerable variation in evolutionary conservation (**Fig. 1a**, and **Supplementary Table 2**). In particular, PQSs in *Tmprss2* mRNAs showed increased probability for RG4 formation compared to that of *Ace2* (**Page 10, line 1-3**). In this regard, we think that *Tmprss2* is a good candidate to exemplify the existence of RG4s in SARS-CoV-2 host factors, as well as its potential function in virus infection. In this manuscript, a combination of biochemical, biophysical, and functional approaches clearly showed the importance of RG4 in *Tmprss2* translation and pseudovirus infection (**Fig. 2**, and **Fig. 3**). Of note, gene-specific RG4 targeting agents/strategies are not available heretofore, thus the exact role of RG4 in TMPRSS2 translation awaits further investigation.

By following the reviewer's suggestion, we have also performed polysome shift analysis to test the effect of PDS on global and *Tmprss2* translation (**Figure below**). PDS treatment resulted in not only an inefficient *Tmprss2* translation, but also a decrease in total mRNA amounts associated with both heavy polysome fractions and light polysome fractions in both H1299 and hACE2-293T cells, indicative of reduced global translation (**Figure below**). The suppressive effect of PDS on global translation is consistent with previous observations in other cell lines (*Nat Commun*, 2021, PMID: 32461552).

Intriguingly, although PDS led to global translation block in cells transfected with hG4mut1 plasmids, it did not obviously alter the translation profile of *Tmprss2* mutant mRNAs, suggesting that RG4s, but not global translation attenuation, may contribute to PDS-mediated *Tmprss2* suppression. These results are now described in **Fig. 2i-k**, and **Supplementary Fig. 2n**.

In addition to these new experimental data, we have inserted a few words into the revised manuscript to discuss the potential role of PDS in genome-wide G4s and the importance of gene-specific RG4 targeting strategies (**Page 23, line 9-14**).

Fig. 2i

Fig. 2j

Fig. 2k

Supplementary Fig. 2n

PMID: 2461552 Fig 2C

5. *Figure 5 and Supplementary Fig. 5. The authors only show 2 healthy and 2 patient lung tissue samples, which is not strong enough sample size to support the findings here. More samples are required with proper statistical test performed.*

Response: Thank the reviewer for this important comment. We totally agree with the reviewer that more COVID-19 samples will greatly strengthen the link between TMPRSS2 expression and SARS-CoV-2 pathogenesis. Actually, we have attempted many ways to obtain more COVID-19 lung tissues, but unfortunately failed due to extraordinary limited sources and strict control of these tissues. In combination with the suggestions regarding this issue from

Reviewer 3 (Point 2), we have modified the manuscript in two aspects to address this concern. On the one hand, we have systemically searched the literature and databases on this subject. Indeed, high levels of TMPRSS2 expression were also observed in the lungs of nine autopsy patients with COVID-19 (*J. Pathol*, 2021, PMID: 32930394), whereas its expression was barely detected in the lungs of individuals without COVID-19 from the HPA database (Human Protein Atlas, <https://www.proteinatlas.org/>) (**Figure below**). In addition, a recent proteome study found that TMPRSS2 was up-regulated at 3-6 hours post SARS-CoV-2 infection in human alveolar type 2 cells (*Mol Cell*, 2020, PMID: 33259812), suggesting a potential link between SARS-CoV-2 and TMPRSS2 expression. In general, these results are consistent with our data, indicating increased TMPRSS2 expression in SARS-CoV-2 pathogenesis. On the other hand, we have moved these immunohistochemistry images to Supplementary Data (**Supplementary Fig. 6a, 6b**) and clearly stated and discussed these limitations in the revised manuscript (**Page 25, line 13-17**). We hope this is considerable and acceptable for the reviewer.

6. *Figure 5. The authors showed TMPRSS2 is induced in lung of patient with COVID-19, but there is no experimental evidence provided by the author that G4 (or addition of G4 stabilizer) can downregulate the expression of TMPRSS2 in patient cells, nor a potential G4 regulator (e.g. protein or helicase) being identified, making this a correlation. More experiments and discussion are needed to substantiate the proposed model.*

Response: We appreciate the reviewer for these constructive suggestions. We totally agree with the Reviewer that determining the effect of RG4s on TMPRSS2 expression in patient cells will greatly strengthen the link between TMPRSS2 expression and SARS-CoV-2 pathogenesis. However, to the best of our knowledge, there is no study that use primary lung cells from COVID-19 patients for experiment thus far, probably due to technological difficulties and tissue source limitation. Thus, it is currently impracticable for us to conduct the suggested experiments. Nevertheless, we have inserted a few words into the DISCUSSION section to discuss the important concern as follows “The authentic SARS-CoV-2 assay, or even primary lung cells from COVID-19 patients if possible, is still required to verify the systematically effect of RG4s on SARS-CoV-2 lifecycle.” (**Page 25, line 7-9**).

We also agree on that it is of importance to discuss the potential RG4 regulators involved in TMPRSS2 expression, as well as COVID-19 pathogenesis, which is also raised by **Reviewer 3 (Point 10)**. Following the suggestion, we have revised the DISCUSSION section as follows: “Given the importance of RG4 for *Tmprss2* translation and SARS-CoV-2 infection, it is of interest to identify the regulator(s) modulating *Tmprss2* RG4. On the one hand, a number of known RG4 helicases, such as DHX9, DHX36, DDX3X, DDX5 and DDX21, were elevated in SARS-CoV-2 infected host cells, suggesting that these helicases may play a role in the regulation of RG4s within the host and/or virus genomes, including *Tmprss2*. On the other hand, certain components of SARS-CoV-2 genome, such as NSP3 and NSP13 that could unwind RG4 structures, may participant in modulating RG4 dynamics in both virus and host genomes during SARS-CoV-2 pathogenesis. Anyway, the precise alternation of TMPRSS2 mRNA and proteins in response to SARS-CoV-2 infection await further investigation.” (**Page 24, line 15-22; Page 25, line 1-2**)

Minor comments

7. In this work, the author characterized the RG4 in SARS-CoV-2 genome at the beginning. However, the presence of RG4s in SARS-CoV-2 genome has been reported before (Reference 25, 26, 27). While the author has cited them, it will be important to point out the difference and similarity in the prediction and in vitro experimental validation among the findings in this work and the previous reports to help the readers.

Response: Thank the reviewer for this important suggestion. Inspired by the thoughtful comments from both reviewers (**Reviewer 3, Point 9**), we have revised the DISCUSSION section as follows: “Consistent with our results, Ji et al identified a metastable RG4 structure formed by PQS-13385 in NSP10. PQS-28903, which was neglected in our study, has been reported to form RG4 structure and inhibit Nucleocapsid expression. Collectively, our present study, together with other recent works, suggest a potential involvement of RG4 in SARS-CoV-2 regulation.” (Page 21, line 21-22; Page 22, line 1-3)

In addition, we have also examined the thermal stability of the RG4 structure formed by PQS-13385 (**Figure below**). CD-melting analysis revealed that the T_m of PQS-13385 was increased in KCl compared to LiCl, but was lower than 3-quartet RG4 structure formed by PQS-675, indicating a metastable RG4 structure. This result is now described **Supplementary Fig. 1c**.

Supplementary Fig. 1c

8. From the native gel in Figure 1d, the PQS-675-Mut only shows the ssRNA band but no RG4 band. However, in Figure 1c, PQS-675-Mut shows enhanced fluorescence with G4 ligand NMM/ThT, and specifically, it has relatively strong fluorescence with ThT. The author needs to address this.

Response: Thank the reviewer for the careful reviewing. Based on various thoughtful suggestions from all three Reviewers, we have utilized multiple approaches, including CD spectrum, NMM probe, FRET, thermal melting measurement, native PAGE, and BG4 staining, to confirm the formation of PQS-675 RG4 structure. Specific to this concern, recent studies suggest that the fluorescence of ThT signal can be enhanced in the presence of RNA molecules, such as poly(A), poly(G) and the homoduplex formed by [TG(GA)₃]_n (*Nucleic Acids Res*, 2015, PMID: 25883145; *Analyst*, 2019, PMID: 31750449). Thus, it is possible that the ThT fluorescence of PQS-675-Mut could result from non-RG4-specific staining. In this regard, we have inserted a few words in the revised manuscript as follows “Despite this decrease, there was strong fluorescence of ThT for PQS-675-Mut (Fig. 1c), consistent with previous observations that ThT staining may have non-specific effects rather than RG4. Therefore, the capability of PQS-675 for RG4 formation was further confirmed by the fluorescence resonance energy transfer (FRET) assay, as evidenced by increased FRET level of this PQS, but not its mutant, in KCl (Fig. 1d).” (**Page 10, line 17-22**)

9. *Figure 2a and 2d: Did the author use the same concentration of RG4 stabilizer for Figure 2a and Figure 2d? Why does the reduced TMPRSS2 protein levels in Figure 2d is much more significant than Figure 2a?*

Response: Thank the reviewer for the critical comment, which is also raised by **Reviewer 2 (Point 3)**. As indicated in FIGURE LEGENDS, Figure 2a and Figure 2d utilized the same dose (10 μM) of RG4 stabilizers, including PDS, cPDS and TMPyP4, to determine their effects on endogenous and exogenous expression of TMPRSS2, respectively. Following this comment, we have carefully examined the original data and re-confirmed these results (**Figure below**), which consistently suggested increased inhibition in Figure 2d. Endogenous TMPRSS2 (Figure 2a) is much lower than exogenous TMPRSS2 (Figure 2d), which might contribute to this difference. Meanwhile, the

differences between endogenous gene expression system and plasmid-mediated exogenous expression, such as the latter producing non-intron RNAs, might also contribute to RG4 stabilizer-induced inhibitory efficiency. Nevertheless, both Figure 2a and Figure 2d showed a significant inhibition of RG4 stabilizers on TMPRSS2 expression.

To improve the reliability and quality of the TMPRSS2 blot in Figure 2a and Figure 2d, we are now updated these images in the revised manuscript. The replicated data of Figure 2a, 2d are shown in **Supplementary Fig. 2a**, and **2c**.

10. Figure 2c: Why there is no TMPRSS2 expression in the pcDNA control (lane 1)? There should be some endogenous TMPRSS2 in H1299 cell.

Response: Thank the reviewer for the careful reviewing. To exclude the potential interference of endogenous TMPRSS2, we used anti-HA-tag antibody, but not anti-TMPRSS2 antibody, to detect the expression of exogenous TMPRSS2. Thus, there is no HA-TMPRSS2 band in the pcDNA control. We apologize for this unclear label in the Figures and have modified the label as “HA-TMPRSS2” instead of “TMPRSS2” in the revised manuscript.

11. Figure 3: The authors should give more details on the design of pseudovirus luciferase system, and it is better to include a scheme.

Response: Thank the reviewer for the suggestion. As suggested, we have provided detailed information of the pseudovirus luciferase system in the RESULT and METHODS section as follows: “The pseudoviruses then infected human lung cells and expressed a fluorescence protein Renilla for quantification” (Page 16, line 6-8), “These SARS-CoV-2-S-luc pseudotyped viral particles consist of a lentiviral core with Renilla element, and the SARS-CoV-2 S glycoprotein on its envelope. Therefore, cells infected with pseudoviruses can be examined via luciferase activity.” (Page 28, line 3-6) In addition, we have added a new graphical figure to explain the pseudovirus system in Fig. 3a, and shown below.

12. Figure 4: The authors should give more details on the design of the VSV-SARS-2-S-luc-infected mouse models.

Response: Thank the reviewer for the comment. The procedures and specific details of the *in vivo* experiment have been described in FIGURE LEGENDS (Fig. 5a) and the METHODS section (Page 28, line 1-13) in our original manuscript. Following the reviewer’s suggestion, we have also added a few words into the revised RESULT section to provide a little bit detail as follows: “In brief, mice were initially inoculated with 5×10^{11} genomic copies (GCs) of AAV9-hACE2, and then infected with 2.5×10^8 relative light units (RLUs) of VSV-SARS-2-S-luc pseudovirus 7 days after.” (Page 18, line 18-20)

13. Page 17. 'Interestingly, these newly characterized host factors also have the potential to adopt RG4 structures (data not shown), indicating a prevalent regulatory role of RG4 in SARS-CoV-2 host factors'. I suggest the authors show these data (e.g. QGRS score, conservation, etc) or cite reference here.

Response: We thank the reviewer for providing this suggestion, which is similarly raised by **Reviewer 3 (Point 7)**. As suggested, the RG4 prediction of *Axl* and *Furin* is now shown in **Supplementary Fig. 7**.

Supplementary Fig. 7

Export to Excel **Axl**

Data View

Search Parameters: QGRS Max Length: 30 | Min G-Group Size: 2 | Loop size: from 0 to 36 | Loop search string:

QGRS sequences found (overlaps not included)

Position	Length	QGRS	G-Score
21	15	GGGGGGAGGGCCGGG	40
94	28	GGCACCCATGGCGTGGCGGTGCCCCAGG	20
125	21	GGCAGGGTCCCCTGGCCTGG	17
166	29	GGCGTGCATGGCCCCAGGGGCACGCAGG	21
236	30	GGTGCCCCGGGACTCACGGGCACCCTTCGG	20
298	21	GGTACATTGGCTTCGGGATGG	18
370	26	GGATGAACAGGATGACTGGATAGTGG	19
491	20	GGCTATGTTGGGCTGGAGGG	16
526	26	GGAGGAGCCCGAAGACAGGACTGTGG	10
640	21	GGCCACGGCTCCAAGTCAACGG	20
817	27	GGAGCTGGAGGGTGGCTTGGACTCCAGG	20
903	19	GGATGGGCATCCAGGCGGG	17
1059	30	GGACCCACTGGCTTCTGTGGAGACGCCGG	20
1131	29	GGAAATGGAGCCAGCCCTTCGTGATTGG	13
1197	27	GGTACCGGTGGCGTATCAAGCCAGG	16
1243	23	GGACATAGGGCTAAGGCAAGAGG	21
1273	18	GGAGCTGCAGGGGACGG	15
1337	15	GGGGATGGACCTGG	16
1369	18	GGAGGCCTGGCGCCAGG	16
1443	25	GGCCCTGGTGGTATGACTGCTAGG	10
1567	25	GGAAAGAGGTGAAGCTGGTAGTCAGG	20
1669	26	GGAGAAGCTGGGATGTGATGGTGG	13
1698	30	GGCACAAAGGTGGCCCTGGGAAGACTCTGG	19
1733	26	GGAGAGTTTGGAGCTGTGATGGAAAGG	14
2113	16	GGACCTGGCGCCAGG	18
2298	23	GGTCCTCGGGGTGACAATGTGG	12
2451	29	GGTGCTGGGAGCTAAATCCCCAGGACCGG	11
2572	15	GGATGAGGGTGGAGG	18
2766	13	GGCAGGAGGATGG	20
2993	12	GGAGGGGTTGG	19
3384	25	GGTTTACCATGTTGGCCAGGCTGG	11
3706	19	GGAGGTTCTAAGGCCTAGG	16
4498	27	GGATGCCTCCTTTCCCGCAGGGGATGG	4
4593	29	GGACTCTGGTGCTCCAGAGGGGCTCAGG	16

Export to Excel **Furin**

Data View

Search Parameters: QGRS Max Length: 30 | Min G-Group Size: 2 | Loop size: from 0 to 36 | Loop search string:

QGRS sequences found (overlaps not included)

Position	Length	QGRS	G-Score
11	29	GGAAAGGGCCCCGCCCTGTGAAAGGGATAGG	10
67	24	GGCGGGGAAGCAGCAGCGCCAGG	14
101	30	GGTGCTCTGGAGCTGGATGGTGAAGGTCGG	19
198	29	GGCAGTGAGCAGGCACTGGAGCCGAGG	18
238	24	GGCCAAAGAGACGGCGCTCCAGG	18
286	25	GGAGCTGAGGCCCTGGTTGCTATGG	18
401	16	GGAGGCCACAGCGTGG	16
427	29	GGCACGGAAAGCATGGTTCTCAACCTGG	13
467	29	GGGGACTATTACCACTTCTGGCATCGAGG	6
567	18	GGCTGGAAACAGCAGGTGG	15
688	27	GGCGGCTTGGCGCAGGGCTACACAGG	20
719	29	GGCATTGTGGTCTCCATTCTGGACGATGG	16
766	27	GGACTTGGCAGGCAATTATGATCTGG	10
861	21	GGCACGGCACACGGTGTGG	19
889	30	GGCTGCGGTGGCCAACAACGGTGTCTGTGG	20
947	28	GGAGGGGTGCGCATGTGGATGGCGAGG	14
1044	30	GGGGCCCGAGGATGACGGCAAGACAGTGG	17
1113	28	GGGTTAGCCAGGGCCGAGGGGGCTGGG	39
1155	24	GGGCCTCGGGGAAACGGGCGCCGG	42
1452	24	GGCGGGACATGCAACACCTGGTGG	9
1516	22	GGCCACCAATGGTGTGGCCGG	16
1556	23	GGCTACGGGCTTTTGGACGCAGG	21
1588	29	GGCCCTGGCCAGAATTGGACCAAGTGG	16
1667	21	GGGAAACGGCTGAGGTGCGG	19
1731	23	GGCTGGAGCAGCCTCAGGCGCGG	13
1899	27	GGGATGAGGATCCCTCTGGCGAGTGGG	18
2065	24	GGCCTGTGTGGTGTGCGAGGAAAGG	16
2371	11	GGAGGTGGAGG	21
2385	20	GGCAACGGCTGCGGGCAGGG	21
2428	12	GGTGGTGGCCGG	20
2518	30	GGGGGTGAAGGTGTACACCATGGACCGTGG	16
2565	27	GGCTGCCCTGAAGCCTGGCAGGAGG	6
2614	22	GGACGAGGGCGGGGGGAGAGG	20
2768	18	GGGAGGCAAGAGGGGTGG	18
2829	30	GGTGGGCCCAAGACCACTGGGGCTGGGG	20
3054	22	GGCAGTCGGGGCTGGCCTAGG	20
3087	22	GGAGGAGCCACTCTCCAAGG	10
3175	29	GGGACCAAGCAAGGCAGGTGCTCCAGG	19
3276	25	GGCCACCAAGGCTGGCGAGCCAAAGG	14
3441	20	GGGTGGTGGTGGGAGGGG	39
3515	22	GGATCTCAGGGGCTGTTGAGG	13
3658	27	GGCTGCCCTGGCCCTGAGGTGTGGGGG	20
3753	27	GGGCTCAAGGAAAGGGGTCCAGTGG	20
3782	30	GGGGCAGGCTGACATCTGTGTTCAAGTGG	1
3825	20	GGGGTTTCATAGTCACTGG	16
3861	13	GGTGGCAGGTGG	20
4103	27	GGCTGGTTTTGTAAGATGCTGGGTTGG	9

14. *In some places in the manuscript, some typos were spotted. For example, 'SRAS-CoV-2' in Page 14 should be SARS-CoV-2.*

Response: We appreciate the reviewer for the careful reviewing. We are so sorry for our carelessness. We have thoroughly checked the manuscript and corrected these errors.

Reviewer 2

This manuscript by Liu et al. assesses the contribution of RNA G-quadruplexes (RG4s) in modulating SARS-CoV-2 infection. The authors mainly focus on characterising the role of a single RG4 motif found within the ORF of the Tmprss2 protease known to regulate SARS-CoV-2 entry and show that stabilisation of this RG4 motif, using small molecules, inhibit Tmprss2 translation and SARS-CoV-2 infection both in vitro and in vivo. This manuscript is timely and may be of great interest for identifying new approaches to tackle SARS-CoV-2 infection. Nevertheless, some crucial experiments and controls are needed to fully support the conclusions of the manuscript. Here are some comments/suggestions:

Response: We are grateful to the reviewer for providing these thoughtful suggestions to improve our study.

Major comments

- 1. The biophysical characterisation of viral and host RG4s is lacking important experiments and controls to fully support RG4 formation and stabilisation by the small molecules. The authors should report the thermal stability of each studied RG4 motif and demonstrate the potassium dependency of RG4 stability. CD spectra are not sufficient to support RG4 formation (for example the CD spectra of PQS-13385 and PQS-24268 are identical while only the former is proposed to fold into RG4). The authors should also study the stability of RG4s (especially PQS-675) in a longer context as it is known that flanking sequences impact greatly both folding and stability. Finally, it is crucial to demonstrate that the three small molecules (PDS, cPDS and TMPyP4) are able to bind to and stabilise PQS-675 but not its mutated version. It is noteworthy that TMPyP4 have been reported for destabilising RG4s and enhance translation (see Morris et al. NAR 2012, 40, 4137-4145 for example).*

Response: Thank the reviewer for these insightful and crucial comments. We have performed the suggested experiments to address these concerns and described the results as the follows:

(1) We totally agree with the reviewer that it is of importance to determine RG4 thermal stability and their relationship with potassium. Similar concern was also raised by **Reviewer 1 (Point 1)**. Following the suggestion, we have verified the thermostability of the RG4 structure in PQS-675 (within human *Tmprss2* mRNA), PQS-1370 (within murine *Tmprss2* mRNA) and PQS-13385 (within SARS-CoV-2 genome) by using FRET- and CD- melting (**Figure below**). It revealed that the T_m of PQS-675 was increased in KCl (150 mM) compared to LiCl. Similar results were also obtained for PQS-1370 and PQS-13385. These results are now described in **Fig. 1d, 1e, 1f, 4c, Supplementary Fig. 1c, h, and Supplementary table 3**.

(2) As suggested, we have determined the influence of flanking sequences in PQS-675 RG4 formation and stability. To this end, we have generated the construct containing four additional nucleotides on either side of the PQS-675 RG4-forming sequence (PQS-675FL). PQS-675FL showed typical positive and negative molar ellipticity peaks at 264 nm and 238 nm, albeit its spectrum intensity and T_m were lower than PQS-675 (**Figure below**),

supporting the capability of this PQS for RG4 formation in a longer context, as well as the potential effect of flanking sequences on its formation and stability. This result is similar with the results reported in previous studies (*FEBS J*, 2009, PMID: 19490117). These data are now described in **Supplementary Fig. 1g**.

Supplementary Fig. 1g

(3) As for the binding and stabilizing of RG4 stabilizers to PQS-675, the **Reviewer 1** has also raised a similar comment (**Point 2**). Given that PDS is one of most popular RG4 stabilizer and predominantly used in our study, as well as these experiments are somewhat costly and time-consuming, we exemplified the effect of PDS on PQS-675 RG4 formation and thermostability. In short, we have performed CD-melting, FRET-melting, and BLI assays to verify the binding and stabilizing of PDS on PQS-675. These results are now described in **Fig. 1d, 1e, 1f, 1h, Supplementary Fig. 1h**, and **Supplementary table 3**. To avoid redundant description, the reviewer may refer to our above response (**Reviewer 1, Point 2**).

Fig. 1h

(4) We fully agree with the reviewer that the effect TMPyP4 on RG4 stability is uncertain. As the reviewer mentioned, it has been reported that TMPyP4 can unwind RG4 (*Nucleic Acid Res*, 2012, PMID: 22266651). However, some studies also reported the stabilizing effect of TMPyP4 on RG4 structures (*Cell Chem Biol*, 2016, PMID: 27617851; *Sci Adv*, 2016, PMID: 27051880). In this study, we found that TMPyP4 inhibited TMPRSS2 translation by stabilizing PQS-675 (**Fig. 2a, 2d and 2g**), indicating TMPyP4 as a RG4 stabilizer for this RG4. Anyway, we have cited the related references and inserted a sentence into the revised manuscript as follows “However, It is noteworthy that TMPyP4 can also destabilize RG4s, thus its effects on GQS-675 should be carefully evaluated with caution in the future.” (**Page 13, line 11-13**)

2. *The detection of RG4 in living cells by immunofluorescence assay is lacking of quantification. The authors should report the quantification of FAM/BG4 foci number after FAM-labelled RNAs transfection. The authors should also demonstrate that an antisense oligonucleotide to the RG4 and treatment with a small molecule stabiliser decrease and increase the colocalisation of the signal respectively.*

Response: Thank the reviewer for these critical comments. Following the suggestion, PDS, as well as antisense oligonucleotides (ASOs) that are complementary to PQS-675 sequences, have been employed in immunofluorescence assay and the results have been quantified (**Figure below**). PDS enhanced colocalization of FAM-RNAs and RG4s in cells transfected with the WT RNAs. Conversely, ASOs reduced this colocalization, in line with the notion that ASOs could unwind RG4 structures (*Nat Chem Biol*, 2014, PMID: 24633353). These results are now described in **Fig. 1i and 1j**.

Fig.1i**Fig.1j**
3. *The impact of small molecules on Tmprss2 translation is somehow inconsistent and the authors need to clarify some of the results. The efficiency of the small molecules seems indeed to be cell- and/or target-specific. For example, PDS has little to no effect on inhibiting the expression of endogenous Tmprss2 in H1299 cells (Fig. 2a), seems to be highly efficient in inhibiting an ectopically expressed Tmprss2 in the same cell line (Fig. 2d) and has no effect on the translation of the same construct in hACE2-293T cells (Fig. 2g). Such discrepancies are observed for cPDS and TMPyP4. It is unclear what is the exact protocol the authors have followed and the number of replicates that have been performed, but the authors should report additional data to support that their findings are robust and reproducible. The authors should report all WB replicates together with the quantification of band intensity in a Supplementary Fig. for the reader to be able to appreciate the variability associated with these experiments.*

Response: Thank the reviewer for the careful reviewing. To address this point that was also raised by the **Reviewer 1 (Point 9)**, we have carefully examined the original data and showed the results of three repeated experiments here

(Red outlines indicating the representative image in the main text). These results indeed confirmed increased inhibitory effect in **Fig. 2d** (**Figure below**). Possible explanations for this difference may include low expression of endogenous TMPRSS2, and the regulation of plasmid-mediated exogenous expression. To avoid redundant description, please refer to our above response to **Reviewer 1 (Point 9)** for more details.

We totally agree with the reviewer that the effects of these RG4 small molecules may be cell- and/or target-specific. Similar cell-specific dependence was also observed in the regulation of pri-miR-26a-1 processing in our previous study (*J Hepatol*, 2020, PMID: 32165252). More importantly, although the inhibitory effect in **Fig. 2g** was less than that of **Fig. 2d** and **Fig. 2a**, statistical analysis based on triple replicates showed a significant reduction of TMPRSS2 by these small molecules. In order to improve the reliability and quality of the TMPRSS2 blot in **Fig. 2a** and **Fig. 2d**, we have now updated these images in the revised manuscript, together with ImageJ quantification. All WB replicates associate with **Fig. 2** are now described in **Supplementary Fig. 2a-d, 2j-k**.

4. *The authors assessed the impact of PDS on the distribution of ribosomes within the Tmprss2 mRNA in H1299 and hACE2-293T cells. Again, the observations in both cell lines are not consistent. While PDS shift the mRNA peak toward monosomes in both cell lines, the authors observed an accumulation of 40 and 60S monosomes in H1299 cells but an accumulation of 80S monosomes in hACE2-293T cells. This point should be clarified. The authors report polysome gradient profiles (A254) only for untreated cells but should report similar profiles after PDS treatment to demonstrate that PDS is not a general inhibitor of translation. Finally, the authors should demonstrate that this shift of Tmprss2 mRNA towards the monosome fraction is RG4-dependent by studying the G4mut-transfected hACE2-293T cells. This latter control is crucial to appreciate the contribution of RG4 to Tmprss2 translation.*

Response: Thank the reviewer for this critical comment, which is also raised by **Reviewer 3 (Point 6)**. We have carefully examined the original data, repeated these experiments in H1299 and hACE2-293T cells and obtained similar and consistent results (Red outlines indicating the original image in the manuscript). These data consistently showed that PDS indeed induced *Tmprss2* mRNA peak shifted to 40S and 60S monosomes in H1299 cells, but

an accumulation of 80S monosomes in hACE2-293T cells (**Figure below**). Similar with the western blot results in **Fig. 2**, it is possible that this discrepancy may be cell type-specific.

Our ribosome profiling experiments clearly showed that PDS indeed inhibited the global translation in both H1299 and hACE2-293T. This phenotype is consistent with previous observations in other cell lines (*Nat Commun*, 2021, PMID: 32461552). Of note, PDS induced translational inhibition of *Tmprss2* was diminished with hG4mut plasmids transfection, albeit the global translation block (**Figure below**), suggesting that RG4s, but not global translation attenuation, may contribute to PDS-mediated *Tmprss2* suppression. These results are now described in **Fig. 2i-k**, and **Supplementary Fig. 2n**. The reviewer may also refer to our above response (**Reviewer 1, Point 4**) for more details.

5. In Fig. 3c, it is surprising that PDS has a similar inhibitory effect in cells transfected with the empty or the G4WT vector. This observation suggests that overexpression of *Tmprss2* does not stimulate virus entry. In order to clarify this point, the authors should perform the same experiment using G4WT- and G4mut-transfected hACE2 293T cells that do not endogenously express *Tmprss2*.

Response: Thank the reviewer for this important comment, which is also raised by **Reviewer 3 (Point 1)**. **Fig. 3b** in the original manuscript (now designated as **Fig. 3c** and **3d** in the revised manuscript) showed that plasmid-mediated TMPRSS2 overexpression can stimulate virus entry. By following the suggestion, we have determined the influence of PQS-675 in virus entry by using the hG4WT and hG4mut1 plasmids in hACE2-293T cells (**Figure below**). In line with the observation in H1299, overexpression of TMPRSS2 induced virus entry in both hG4WT and hG4mut1 cells. Moreover, PDS treatment reduced pseudovirus entry in hG4WT cells, and this inhibition was abolished in hG4mut1 cells, indicating the inhibitory effect of PQS-675 on virus infection. This result is now described in **Fig. 3f**.

Fig. 3f

6. A major caveat with the *in vivo* experiments is that the PQS-675 RG4 found within the human *Tmprss2* is not present in the corresponding murine mRNA. The authors then postulate that all their findings related to the human *Tmprss2* mRNA are effective for the murine *Tmprss2* mRNA. This

hypothesis must carefully be validated by reproducing key experiments with the murine mRNA. Experiments, such as the ones described in Fig. 2 and 3, should be reproduced with vectors encoding the murine Tmprss2 comprising WT and mutated RG4 motifs. Similarly, the authors should assess the impact of PDS on the translation of the WT and mutated human Tmprss2 when heterogeneously expressed in mice. These experiments are crucial to support the role of RG4s in modulating SARS-CoV-2 entry in vivo.

Response: We thank the reviewer for these thoughtful suggestions. As suggested, we have assessed the effect of RG4 on murine TMPRSS2 expression. RG4 stabilizers were capable to reduce endogenous TMPRSS2 expression in murine lewis lung carcinoma cells (LLC). By generating the murine *Tmprss2* (mG4WT) and its G4-mutant (mG4mut) plasmids, we further showed that the level of exogenous TMPRSS2 was higher in mG4mut cells than mG4WT cells. Moreover, RG4 stabilizers diminished the induction of TMPRSS2 by mG4WT plasmids, but not by mG4mut plasmids, further supporting the inhibitory role of RG4 in murine TMPRSS2 expression. In addition, the human *Tmprss2* and its RG4 mutants were heterogeneously expressed in LLC cells, respectively, and the impact of PDS on their expression was assessed. All three of the RG4-mutant plasmids (hG4mut1/2/3) led to an increase in human TMPRSS2 expression compared to the hG4WT plasmids (The various types of mutation aim to exclude the potential interference of codon usage bias as suggested by **Reviewer 1 Point 3**). More importantly, heterogeneous expression of human *Tmprss2*, but not its RG4 mutants, was repressed by RG4 stabilizers. These results are now described in **Fig. 4d-i, Supplementary Fig. 4d, e** and **shown below**.

Fig. 4d

Fig. 4e

Fig. 4f

Fig. 4h

Fig. 4g

Fig. 4i

Supplementary Fig. 2i

Supplementary Fig. 4d

Supplementary Fig. 4e

7. Differences in *Tmprss2* protein levels (saline vs PDS) on the Western blot reported in Fig. 4i are difficult to appreciate. The authors should quantify the intensity of the bands and perform a statistical test to assess the relevance of their result. Because this result is central to this manuscript, the author should consider quantifying the difference in protein level using a more quantitative technique such as ELISA.

Response: We sincerely thank the reviewer for providing this suggestion. Following the suggestion, we have added the quantification of western blot data in the revised manuscript (**Fig. 5h**), which showed the repression role of PDS on murine TMPRSS2 expression. We have also performed the suggested ELISA assay, which confirmed an increase in TMPRSS2 protein level in lungs of PDS treated mice (**Fig. 5f**), consistent with the western blot data (**Fig. 5h**). These results are now described **Fig. 5h**, and **5f**.

8. *Because RG4s are found within the virus genome (S protein) and within the human ACE2 receptor, the authors need to assess whether PDS affect the transcription or translation of both viral and host factors to demonstrate that the observed decrease in SARS-CoV-2 infection in vivo is solely due to the inhibition of Tmprss2 translation. Without these data, the current observations are purely descriptive and not causative.*

Response: Thank the reviewer for this insightful comment. Based on our results showing the effect of PDS on global translation (**Fig. 2i-k**), as well as numerous RG4s in the host and virus genome, it is likely that the observed decrease in SARS-CoV-2 pseudovirus infection by PDS treatment may result from its effects on multiple RG4s, instead of *Tmprss2* solely. As mentioned in the manuscript for several times, this study aims to identify the biological, physiological and pathological functions of RG4 in gene regulation and SARS-CoV-2 infection by exemplifying *Tmprss2* (**Page 13, line 2-3; Page 21, line 4-6**). Meanwhile, the universal role of PDS on multiple potential targets and SARS-CoV-2 infection has been extensively discussed in the DISCUSSION

section in the revised manuscript (**Page 23, line 9-14**). Of note, our results showed that TMPRSS2 overexpression can increase SARS-CoV-2 infection (**Fig. 3c-f**). More intriguingly, PDS was sufficient to abolish the increase of SARS-CoV-2 infection in cells with exogenous TMPRSS2 overexpression, but did not affect SARS-CoV-2 infection induced by G4-mut TMPRSS2 overexpression (**Fig. 3e and 3f**). These data collectively suggest an important contribution of *Tmprss2* RG4 in PDS-mediated inhibition on virus infection.

We fully agree with the Reviewer that it is important to assess the effect of PDS on the transcription or translation of both viral and host factors. However, the SARS-CoV-2-S-luc pseudotyped viral particles used in our study cannot express any of the SARS-CoV-2 genes, but only enveloped with the SARS-CoV-2 S glycoprotein. Accordingly, the effect of PDS on SARS-CoV-2 genome was precluded in our *in vivo* study. For the host factors, the nucleotide sequence of *Ace2* in AAV9-hACE2 has been mostly transformed by using codon optimization, thus deviated from the original sequences. In this regard, it is impossible to examine the effect of PDS on ACE2 expression. During the preparation of this manuscript, we have indeed noticed some other host factors which may also regulated by RG4 structures, such as *Axl* and *Furin* (**Supplementary Fig. 7**). Indeed, our preliminary data suggest that the expression of AXL can be inhibited by PDS (**Figure below**). Because this work is still in progress in our lab, and is possibly beyond the scope of the present study, we thus did not present this result in the manuscript.

The RG4 prediction of *Axl* and *Furin* is now shown in **Supplementary Fig. 7**. In addition, this constructive comment is a valid point for which we have add some sentences to discuss other RG4 structures related to SARS-CoV-2 infection in the DISCUSSION section (**Page 22, line 15-19**). We hope this is acceptable for the reviewer.

Supplementary Fig. 7

Export to Excel **Axl**

Data View

Search Parameters: QGRS Max Length: 30 | Min G-Group Size: 2 | Loop size: from 0 to 36 | Loop search string:

QGRS sequences found (overlaps not included)

Position	Length	QGRS	G-Score
21	15	GGGGGGAAGCCGGG	40
94	28	GGCACCCATGGCGTGGGGTGCCCAAG	20
125	21	GGCAGGGTCCCGCTGGCTGG	17
166	29	GGCGTGCATGGCCCCAGGGGACGCAGG	21
236	30	GGTGCCCGGGACTCACGGCACCCCTCGG	20
298	21	GGTACATTGGCTTCGGGATGG	18
370	26	GGATGAACAAGGATGACTGGATAGTGG	19
491	20	GGCTATGTTGGGCTGGAGGG	16
526	26	GGAGGAGCCCGAAGACAAGACTGTGG	10
640	21	GGCCACGGCTCCAGGTCACGG	20
817	27	GGAGCTGGAAGTGGCTGGACTCCAAG	20
903	19	GGATGGCATCCAAGCGGG	17
1059	30	GGACCCACTGGCTTCTGTGGAGACGCCGG	20
1131	29	GGAAAGGGAGCCAGGCTTCTGTGATTGG	13
1197	27	GGTACCGGGTGGCGTATCAAGCCAGG	16
1243	23	GGACATAAGGCTAAAGCAAGAAG	21
1273	18	GGAGCTGCAAGGGGACCG	15
1337	15	GGGGATGGACCTGG	16
1369	18	GGAGGCTGGCGCCAGG	16
1443	25	GGCCCTGGTGGTATGTACTGCTAGG	10
1567	25	GGAAAGAGGTGAAGTGGTAGTCAGG	20
1669	26	GGAGAAGCTGGGGATGTGATGGTGG	13
1698	30	GGCACAAGTGGCCCTGGGGAAGACTCTGG	19
1733	26	GGAGAGTTTGGAGCTGTGATGGAAAGG	14
2113	16	GGACCTGGCGCCAGG	18
2298	23	GGTCCCTGGGGTGACAATGTGG	12
2451	29	GGTGCTGGAGCTAAATCCCCAGGACCCGG	11
2572	15	GGATGAGGGTGGAGGG	18
2766	13	GGCAGGAGGATGG	20
2993	12	GGAAAGGGTTGG	19
3384	25	GGTTTCACCATGTTGGCCAGGCTGG	11
3706	19	GGAGGTTCTAAGGCCTAGG	16
4498	27	GGATGCCTCCTTCCCGAGGGGATGG	4
4593	29	GGACTCTGGTGCCCTCAGAGGGGCTCAGG	16

Expression of TMPRSS2 in lungs of PDS treated mice. (data not shown in the manuscript)

Export to Excel **Furin**

Data View

Search Parameters: QGRS Max Length: 30 | Min G-Group Size: 2 | Loop size: from 0 to 36 | Loop search string:

QGRS sequences found (overlaps not included)

Position	Length	QGRS	G-Score
11	29	GGAAAGGGCCCCGCCCTGTGAAAGGATAGG	10
67	24	GGCGGGGAAGCAGCAGCGGCCAAG	14
101	30	GGTGCTCTGGAGCTGGATGGTGAAGGTCGG	19
198	29	GGCAGTGAGCAAGCACCTGGAGCCGAGG	18
238	24	GGCCAAAGAGACGGCGCTCCAGG	18
286	25	GGAGCTGAGGCCCTGGTTGCTATGG	18
401	16	GGAGGCCAGCGGTTGG	16
427	29	GGCACGGAAAGCATGGTTCTCAACCTGG	13
467	29	GGGGACTATTACCACTTCTGCATCGAAG	6
567	18	GGCTGGAAACAGCAGGTGG	15
688	27	GGCGGCTGGCGCAGGGCTACACAGG	20
719	29	GGCATTGTGGTCTCCATTCTGGACGATGG	16
766	27	GGACTTGGCAGGCAATTATGATCCTGG	10
861	21	GGCACGGCACAGGTTGTGCGG	19
889	30	GGCTCGGGTGGCAAACAACGGTGTCTGTGG	20
947	28	GGAGGGTGGCGCATGCTGGATGGCGAAG	14
1044	30	GGGGCCCCGAGGATGACGGCAAGACAGTGG	17
1113	28	GGTTAGCCAGGGCCGAGGGGGCTGGG	39
1155	24	GGGCTCTGGGGAACGGGGCCGGG	42
1452	24	GGCGGGACATGCAACACCTGGTGG	9
1516	22	GGCCACCAATGGTGTGGGCCGG	16
1556	23	GGCTACGGGCTTTGGACGCAAG	21
1588	29	GGCCCTGGCCAGAATTGGACCACAGTGG	16
1667	21	GGGAAACGGCTCGAGGTTGCGG	19
1731	23	GGCTGGAGCAGCTCAAGGCGCGG	13
1899	27	GGGATGAGGATCCCTCTGGCGAGTGGG	18
2065	24	GGCCTGTGTGGTGTGCGGAAAGG	16
2371	11	GGAGGTGGAGG	21
2385	20	GGCAACGGCTGGCGGCGAGGG	20
2428	12	GGTGGTGGCCGG	20
2518	30	GGGGGTGAAAGGTGTACACCATGGACCGTGG	16
2565	27	GGCTGCCCCGAAAGCCTGGCAGGAGG	6
2614	22	GGACGAGGGCCGGGGCGAGAGG	20
2768	18	GGGAGGCAAGAGGGGTTGG	18
2829	30	GGTGGGCCCAAGACCAGCTGGGGCGTGGGG	20
3054	22	GGCAGTCGGGGCTGGCCTAGG	20
3087	22	GGAGGAGGCCACCTCTCCAAGG	10
3175	29	GGGACCAAAGCAAGGCAAGTGCCTCCAGG	19
3276	25	GGCCACCAAGGCTGGCGACGCCAAAG	14
3441	20	GGGTGGGTGGTGGGAGGGG	39
3515	22	GGATCTCAGGGGCTGTGTTGAGG	13
3658	27	GGCTGCCCTGGCCCTGAGGTTGGGGGG	20
3753	27	GGGCTCAAAGGAAAGGGGTTCCAGTGG	20
3782	30	GGGGCAGGCTGACATCTGTGTTCAAGTGG	1
3825	20	GGGGGTTCTAAGGCTACTGG	16
3861	13	GGTGGGCAAGTGG	20
4103	27	GGCTGGTTTTGTAAAGATGCTGGGTTGG	9

9. The authors wrote in the discussion section: “Thus, our present and previous studies strongly suggest the effectiveness and safety of PDS in vivo”. This statement is surprising as the acute toxicity of PDS and its ability to trigger genetic instability is well documented. To support their claim the authors should report the impact of PDS treatment on mice fitness and lifespan. The clinical and hematological parameters reported in Supplementary Fig. 4e. are not very informative.

Response: Thank the reviewer for this important comment. We agree with reviewer that the safety and toxicity of PDS still remain controversial. For

instance, PDS was thought to cause double-strand breaks in some studies (Aging, 2017, PMID: 28904242), whereas in some others it was found to even mildly mitigate the formation of such DNA damage (iScience, 2019, PMID: 31678912). To address the reviewer's concern, we have supplemented the body weight changes (**Figure below**). In the toxicity experiment, there were no mice died during nine days continued injection of low dose (6 mg/kg) or high dose (30 mg/kg) PDS, and no abnormal behaviors or body weight loss were observed in the low dose group. Relatively, body weight was transiently reduced in the high dose group, as it went back to normal 8 days after drug withdrawal. These data revealed low toxicity of PDS in mice and are now described **Supplementary Fig. 5a**.

However, we recognize that safety is an important issue, which should be declared cautiously. In this regard, we revised the DISCUSSION section as follows: "In addition, here we provided some evidence on the low toxicity of PDS in mouse, however, its druggability and clinical utility should be carefully evaluated in the future." (Page 23, line 14-16)

Supplementary Fig. 5a

Minor comments

10. The authors may want to consider changing the title of their manuscript to reflect that their work mainly focuses on *Tmprss2* mRNA translation

inhibition rather than targeting the SARS-CoV-2 viral genome. As it stands the title may be misleading.

Response: We are grateful to reviewer for pointing out this issue, and we have revised this title to “RNA G-quadruplex in TMPRSS2 reduces SARS-CoV-2 infection”. We hope this is appropriate.

11. *When describing sequence mutants, the authors should refer to the nucleobases (i.e. guanine and adenine) rather than the corresponding monophosphate (guanylic and adenylic acids).*

Response: Thank the reviewer for this suggestion. We have modified them in our revised manuscript.

12. *In the introduction, the authors report that the UK variant is about “70% more transmittable”, but this point hasn’t been formally demonstrated in the population. This should be clarified.*

Response: We apologize for our unclear sentence, which we have now revised the INTRODUCTION section as the follows “the fast-evolved variation of SARS-CoV-2, which is more transmissible and spreading globally compared to the ancestral virus.” **(Page 4, line 8-9)**

13. *The authors should avoid the use of words such as “extraordinary”, “remarkably” and other superlatives.*

Response: Thank the reviewer for this suggestion. As suggested, we have carefully checked the manuscript and deleted these words.

Reviewer 3

The manuscript by Liu et al. reports the role of RG4 in SARS-CoV-2 infection. Initially, the authors predicted G4-forming sequences in SARS-CoV-2 genome and major host factors for viral entry by bioinformatic analysis, and experimentally confirmed the existence of RG4 within SARS-CoV-2 genome in vitro. Next, the existence and function of RG4 in host factors regulating SARS-CoV-2 infection was exemplified by a canonical RG4 within Tmprss2. Combining several chemical, biochemical, and molecular methods, the authors provided several lines of clear and solid evidence to demonstrate that this RG4 can be formed and inhibit Tmprss2 translation in vitro and in vivo. Moreover, they demonstrated that this RG4 in Tmprss2 can inhibit SARS-CoV-2 entry in cells and reduce SARS-CoV-2 infection in mice by using the pseudovirus system. In addition, the authors found a reduction of Tmprss2 protein in the lung of COVID-19 patients.

This work provides the first evidence to demonstrate the role of RG4 in SARS-CoV-2 infection in vitro and in vivo by using the pseudovirus system. The study is well done, and the data are novel and solid. The written is clear and the flow is excellent. Overall, this timely and highly significant study could provide new ideas against the ongoing COVID-19 pandemic. Nevertheless, there are few concerns that the authors need to address for the manuscript to warrant publication

Response: We are very grateful to the reviewer's encouraging and thoughtful comments.

Major comments

- 1. The data presented in Fig. 3c clearly showed that PDS treatment impairs SARS-CoV-2 infection in H1299 cells transfected with G4WT plasmids, but not with G4mut plasmids. This data is critical to support that the inhibition of RG4 formation within Tmprss2 can suppress SARS-CoV-2 infection. In*

this regard, it is recommended to perform the similar experiment in an additional cell line, such as hACE2-293T, to exclude the potential cell-specific effect and thus strengthen the conclusion.

Response: We sincerely thank the reviewer for this critical suggestion. To address this point that was also raised by the **Reviewer 2 (Point 5)**, we have performed similar virus entry experiments in hACE2-293T cell and obtained coincident results as in H1299 (**Figure below**). Please see also response to **Reviewer 2, Point 5** for details. This result is now described in **Fig. 3f**.

Fig. 3f

2. *The authors used IHC analysis and found that Tmprss2 protein is reduced in the lung of two COVID-19 patients compared with two healthy controls (Figure 5). The sample size was small. It is understandable since the lung tissue of COVID-19 patients is very difficult to obtain and Tmprss2 mRNA level is impracticable to be determined in paraffin-embedded tissues. However, the authors should, at least, clearly state and discuss these limitations in the text, if no more lung tissues of COVID-19 patients can be obtained.*

Response: We thank the reviewer for this important comment, appreciate for understanding our difficult situation on COVID-19 sample collection, and apologize for the insufficient sample size in our study. To address this point which was also raised by the **Reviewer 1 (Point 5)**, we investigated TMPRSS2 levels by searching online database and previous literatures, and provided some theoretical support for our notion. Please see also response to **Reviewer 1, Point 5** for details of description. However, we still recognize that the

conclusion of this argument is not rigorous enough. We thus have moved these immunohistochemistry images to **Supplementary Fig. 6a, 6b**, and clearly stated and discussed these limitations in the revised manuscript (**Page 25, line 13-17**). We hope this is considerable and acceptable for the reviewer.

3. *The authors showed that PDS treatment can decrease exogenous *Tmprss2* translation in G4WT-transfected hACE2-293T cells by using polysome shift analysis (Figure 2j). To further confirm that this effect depends on the G4 sequence, the authors should perform the similar experiment in hACE2-293T cells transfected G4mut plasmids.*

Response: We thank the reviewer for this constructive comment. Inspired by the thoughtful comments from both reviewers (**Reviewer 2, Point 4**), we have analyzed the ribosomal distribution profiles of exogenous *Tmprss2* translation in hG4mut1-transfected hACE2-293T cells. PDS induced translational inhibition of *Tmprss2* was greatly diminished with hG4mut1 plasmids transfection, albeit the global translation block (**Figure below**). These results are now described in **Fig. 2i-k**. To avoid redundant description, the reviewer may refer to our above response to **Reviewer 2, Point 4**.

Fig. 2j

Fig. 2k

Minor comments

4. The quantification of some representative results, such as Fig 1e, Fig 2 (western blots), and Fig 4i, should also be shown.

Response: Thank the reviewer for the careful reviewing. Following the suggestion, the quantification data of immunofluorescence and western blots have been shown in the revised manuscript (Fig 1j, 2a-h, 5h and Supplementary Fig. 5e).

5. The authors showed that the PQS-675 can repress *Tmprss2* translation and function. The sequence of PQS-675 appears to be evolutionarily conserved, which is summarized in Figure 1a. However, it is better to also present the conservation of this PQS-675 by sequence alignment among different species, which can be included in the Supplementary Data.

Response: Thank the reviewer for the suggestion. As suggested, we have supplemented sequence alignment into Supplementary Fig 1e in the revised manuscript.

Supplementary Fig. 1e

6. By using polysome shift analysis, the authors found that PDS treatment resulted in a decrease in endogenous *Tmprss2* mRNA in H1299 cells (Figure 2i), as well as in exogenous *Tmprss2* mRNA in hACE2-293T cells

(Figure 2j). However, the effect of PDS on the percentage of *Tmprss2* mRNA in these two cell lines seems to be different. What is the explanation?

Response: Thank the reviewer for this critical comment, which was also discussed in **Reviewer 2, Point 4**. We have carefully examined the original data, repeated these experiments in H1299 and hACE2-293T cells and obtained consistent results (**Figure below**). We suspected that this discrepancy may be cell type-specific. To avoid redundant description, the reviewer may refer to our above response to **Reviewer 2, Point 4**.

7. As mentioned by the authors, some newly characterized host factors also have the potential to adopt RG4 structures (data not shown). Given that the broad effect of PDS on RG4 structures, it is better to appropriately show this prediction data in Supplemental Data.

Response: We thank the reviewer for this suggestion, which was similarly raised by **Reviewer 1 (Point 13)**. Following the suggestion, the RG4 prediction of Axl and Furin are now shown in **Supplementary Fig. 7**.

Supplementary Fig. 7

Export to Excel **Axl**

Data View

Search Parameters: QGRS Max Length: 30 | Min G-Group Size: 2 | Loop size: from 0 to 36 | Loop search string:

QGRS sequences found (overlaps not included)

Position	Length	QGRS	G-Score
21	15	GGGGGGAGGCCGGG	40
94	28	GGACCCATGGCGTGCGGTGCCAAGG	20
125	21	GGCAGGCTCCCGCTGGCCTGG	17
166	29	GGCGTGATGGCCCCAGGGCAGCAGG	21
236	30	GGTGCCCGGGACTCACGGCACCTTCGG	20
298	21	GGTACATTGGCTTCGGGATGG	18
370	26	GGATGAACAGGATGACTGGATAGTGG	19
491	20	GGCTATGTTGGGCTGGAGGG	16
526	26	GGAGGAGCCGAAGACAGGACTGTGG	10
640	21	GGCCACGGCTCCAGGTCACGG	20
817	27	GGAGCTGGAAGGTGGCTGGACTCCAAG	20
903	19	GGATGGGATCCAGGCGGG	17
1059	30	GGACCCACTGGCTTCCTGTGGAGACGCCGG	20
1131	29	GGAAATGGAGCCAAGCCTTCGTGATTGG	13
1197	27	GGTACCGGCTGGCGTATCAAGCCAGG	16
1243	23	GGACATAGGGTAAGCAAGAGG	21
1273	18	GGAGCTGCAAGGGGACGG	15
1337	15	GGGGATGGACCTGG	16
1369	18	GGAGGCGTGGCGCCAGG	16
1443	25	GGCCCTGGTGTATGTACTGCTAGG	10
1567	25	GGAAAGAGGTGAAGCTGGTAGTCAGG	20
1669	26	GGAGAAGCTGGGGATGTGATGGTGG	13
1698	30	GGCACAAGGTGGCCCTGGGAAGACTCTGG	19
1733	26	GGAGAGTTTGGAGCTGTGATGGAAGG	14
2113	16	GGACCTGGCGGCCAGG	18
2298	23	GGTCCTTCGGGTGACAATGTGG	12
2451	29	GGTGCTGGGAGCTAAATCCCAGGACCGG	11
2572	15	GGATGAGGGTGAGG	18
2766	13	GGCAGGAGGATGG	20
2993	12	GGAAAGGGTTGG	19
3384	25	GGTTTACCATTGTTGGCCAGGCTGG	11
3706	19	GGAGGTTCTAAGCCTAGG	16
4498	27	GGATGCTCCTTCCCGCAGGGGATGG	4
4593	29	GGACTCTGGTGCCTCAGAGGGGCTCAGG	16

Export to Excel **Furin**

Data View

Search Parameters: QGRS Max Length: 30 | Min G-Group Size: 2 | Loop size: from 0 to 36 | Loop search string:

QGRS sequences found (overlaps not included)

Position	Length	QGRS	G-Score
11	29	GGAAAGGGCCCCGCCCTGTGAAGGGGATAAGG	10
67	24	GGCGGGGAAGCAGCAGCGGCCAGG	14
101	30	GGTGCTCTGGAGCTGGATGGTGAAGGTCGG	19
198	29	GGCAGTGAGCAGGCACCTGGGAGCCGAGG	18
238	24	GGCCAAGGAGACGGGCGCTCCAAGG	18
286	25	GGAGCTGAGGCGCTGGTTGCTATGG	18
401	16	GGAGGCCAGCGGTGG	16
427	29	GGCACGGAAAGCATGGTTCTCAACCTGG	13
467	29	GGGGACTATTACACTTCTGGCATCGAGG	6
567	18	GGCTGGAACAGCAGGTGG	15
688	27	GGCGGCGTGGGCGCAGGCTACACAGG	20
719	29	GGCATTGGTCTCCATTCTGGACGATGG	16
766	27	GGACTTGGCAGGCAATTATGATCTCTGG	10
861	21	GGCACGGCACAGCAATTGGCTGG	19
889	30	GGCTGGGTTGGCCAACAGGTTGCTGTGG	20
947	28	GGAGGGTGGCGATGCTGGATGGCGAGG	17
1044	30	GGGGCCCCGAGGATGACGGCAAGCAGTGG	14
1113	28	GGGTTAGCCAAGGCGGAGGGGGCTGGG	39
1155	24	GGGCTCCTGGGGAACGGGGGCCGGG	42
1452	24	GGCGGGACATGCAACACCTGGTGG	9
1516	22	GGCCACCAATGGTGTGGGCCGG	16
1556	23	GGCTACGGGCTTTTGGACGCGG	21
1588	29	GGCCCTGGCCAGAATTGGACCAAGTGG	16
1667	21	GGGAAACGGCTCGAGGTGGCG	19
1731	23	GGCTGGAGCAGCTCAGGCGCGG	13
1899	27	GGGATGAGGATCCCTCTGGCGAGTGGG	18
2065	24	GGCCTGTGTGGTGTGGGGAAGG	16
2371	11	GGAGTGGAGG	21
2385	20	GGCAACGGCTGGCGGCGAGG	21
2428	12	GGTGGTGGCCGG	20
2518	30	GGGGGTGAAGGTGTACACCATGGACCGTGG	16
2565	27	GGCTGCCCTGGAAGCCTGGCAGGAGG	6
2614	22	GGACGAGGGCCGGGCGAGAGG	20
2768	18	GGGAGGCAAGAGGGTGG	18
2829	30	GGTGGGCCAGGACCAAGTGGGGCTGGGG	20
3054	22	GGCAGTCGGGGGCTGGCCTAGG	20
3087	22	GGAGGAGGCCACCTCTCAAAGG	10
3175	29	GGGACCAAGGCAAGGCAAGTGCCTCCAAGG	19
3276	25	GGCCACCAAGGCTGGCGCAGCCAAGG	14
3441	20	GGGTGGTGGTGGGAGGGG	39
3515	22	GGATCTCAGGGGCTGTTTGAAGG	13
3658	27	GGCTGCCCTGGCCCTGAGGTGTGGGGG	20
3753	27	GGGCTCAAGGAAAGGGGTCCAGTGG	20
3782	30	GGGGCAGGCTGACATCTGTGTTTCAAGTGG	1
3825	20	GGGGTTTCATAGGTCACCTGG	16
3861	13	GGTGGGCAAGTGG	20
4103	27	GGCTGGTTTTGTAAAGTGGGTTGG	9

8. This study showed that the RG4 stabilizer PDS can reduce SARS-CoV-2 infection in mouse models. The authors are encouraged to briefly discuss the potential of PDS, as well as other RG4 stabilizers, in translational medicine and its clinical implication. Additionally, G4-forming sequences within SARS-CoV-2 genome have been predicted by this study and other recent reports. Currently, some SARS-CoV-2 variants are emerging globally. It is of interest to discuss the possible implication of RG4 on SARS-CoV-2 variants, if applicable.

Response: We are grateful to the reviewer for providing these thoughtful suggestions to improve our study. Following the suggestion, and combined with our present results, we have revised the DISCUSSION as follows: “Of note, in principle, PDS can bind numerous G4s in the host and virus without target specificity, thus new strategies targeting Tmprss2 RG4 is needed to clarify this regulation in SARS-CoV-2 infection. Unfortunately, there is still no clinical trials on agents based on RG4 mechanism yet. Thus, it is of importance for future investigation to develop gene-specific RG4-targeting agents/strategies. In addition, here we provided some evidence on the low toxicity of PDS in mouse, however, its druggability and clinical utility should be carefully evaluated in the future. Finally, given that both the SARS-CoV-2 and its host factors contain numerous putative RG4s, it is highly anticipated that G4-targeting agents are also able to inhibit the infection of some SARS-CoV-2 variants, if not all.” (Page 23, line 9-19)

9. *As mentioned by the authors, the existence of RG4 within SARS-CoV-2 genome has been reported recently (Ref No. 25-27 in the main text). In this regard, the authors are encouraged to describe the differences between this study and previous literatures, which may be helpful to clarify the novelty and importance of their work, as well as to help the readers for understanding the topic.*

Response: We thank reviewer for raising this concern. Inspired by the thoughtful comments from both reviewers (**Reviewer 1, point 7**), we have examined the thermal stability of the RG4 structures formed by SARS-CoV-2 genome, and discussed the similarities and differences between our study and previous literatures. Please see also response to **Reviewer 1, Point 7** for details.

10. *Numerous RNA binding proteins (RBPs) have been shown to stabilize or unwind RG4 structures. The identification of potential RBPs responsible for the dynamics of RG4s within SARS-CoV-2 genome and host factors,*

particularly Tmprss2, requires intense investigation and may be beyond the scope of this study, however the authors are encouraged to appropriately introduce and discuss this interesting topic, which could throw new insights on further investigation.

Response: Thanks for the suggestion, which is also discussed in **Reviewer 1, Point 6**. We totally agree with the reviewer that it will be important to discuss the role of RG4 binding proteins in COVID-19. Following the suggestion, we have revised the DISCUSSION section to discuss the potential RG4 binding proteins in COVID-19 (**Page 24, line 15-22; Page 25, line 1-2**). For details, please also see our response to **Reviewer 1, Point 6**.

11. The abbreviation of "PQS", suggesting instead of "GQS".

Response: Thank the reviewer for the suggestion. We have replaced "PQS-675" to "GQS-675" in the revised manuscript.

12. It is not clear about G4 when it was first presented, clarify.

Response: We apologize for being unclear. We have corrected it in our revised manuscript.

13. The last sentence of introduction need to be modified.

Response: Thank the reviewer for the careful reviewing. Following the suggestion, we have modified the sentence in the revised manuscript.

REVIEWERS' COMMENTS

Reviewer #1 (Remarks to the Author):

I have thoroughly gone through the revised manuscript and rebuttal file. The authors have sufficiently addressed and clarified all the comments. The revised manuscript is much more improved, and the current data are supportive of the concepts presented and claims made in this manuscript. We commend the authors for their efforts in performing all the additional experiments. I only have 2 minor comments for the authors to address. I recommend the manuscript to be accepted once the authors made the changes.

1. Figure 1h. The authors should indicate what are KD, Ka, Kd in figure legend or manuscript.
2. Figure 2c-h. The authors have modified the label as "HA-TMPRSS2" instead of "TMPRSS2". The corresponding figure legends should also be edited.

Reviewer #2 (Remarks to the Author):

In the revised version of their manuscript, Liu et al. have provided additional experiments that strengthen their original observations and updated their manuscript accordingly. The authors have answered all my comments and took in consideration my main suggestions. I would then recommend the current manuscript for publication in Nat. Commun.

Reviewer #3 (Remarks to the Author):

The authors have satisfactorily addressed all my concerns and revised the manuscript accordingly. Therefore, I recommend publication of this revised manuscript.

Dear Editors and Reviewers,

We are very grateful to the Editor for handling our manuscript and appreciate very much for Reviewers for their constructive suggestions and positive comments that guided us to this new improved version. In response to the remaining minor suggestions, we have addressed the reviewer's requests in the revised manuscript (underlined) and provided a point by-point response below.

Reviewer 1

I have thoroughly gone through the revised manuscript and rebuttal file. The authors have sufficiently addressed and clarified all the comments. The revised manuscript is much more improved, and the current data are supportive of the concepts presented and claims made in this manuscript. We commend the authors for their efforts in performing all the additional experiments. I only have 2 minor comments for the authors to address. I recommend the manuscript to be accepted once the authors made the changes.

Response: We appreciate the reviewer for acknowledging our efforts and providing constructive suggestions.

1. *Figure 1h. The authors should indicate what are KD, Ka, Kd in figure legend or manuscript.*

Response: Thanks for the comment. We have added the suggested information in the FIGURE LEGEND.

2. *Figure 2c-h. The authors have modified the label as "HA-TMPRSS2" instead of "TMPRSS2". The corresponding figure legends should also be edited.*

Response: Thank the reviewer for this suggestion. We have now revised the manuscript as suggested in FIGURE LEGEND for Fig. 2c-h, Fig. 4f-i, Supplementary Fig. 2f-m and Supplementary Fig. 4d-e.

Reviewer 2

In the revised version of their manuscript, Liu et al. have provided additional experiments that strengthen their original observations and updated their manuscript accordingly. The authors have answered all my comments and took in consideration my main suggestions. I would then recommend the current manuscript for publication in Nat. Commun.

Response: We appreciate the reviewer for favorable consideration.

Reviewer 3

The authors have satisfactorily addressed all my concerns and revised the manuscript accordingly. Therefore, I recommend publication of this revised manuscript.

Response: We are grateful to the reviewer for the positive comments and help in improving our manuscript.